# PAWS: PREFERENCE LEARNING WITH ADVANTAGE-WEIGHTED SEGMENTS

## ABSTRACT

Training agents that align with human intentions is a central challenge in machine learning. Preference-based reinforcement learning (PbRL) has emerged as a promising paradigm by leveraging human feedback in the form of trajectory-level comparisons, thereby avoiding the need for explicit reward design or expert demonstrations. However, existing PbRL methods typically rely on per-step reward assignments inferred from trajectory preferences, which introduces inconsistencies and exacerbates the temporal credit assignment problem. In this work, we analyze this issue and demonstrate its adverse impact on policy learning. To address this problem, we propose Preference Learning with Advantage-weighted Segments (PAWS), a novel segment-based preference learning method that updates policies directly with segment-level advantage functions. By preserving segment-level preference information, PAWS ensures stable policy updates while avoiding misleading per-step reward signals. Empirical evaluations on a diverse set of simulated robot manipulation tasks, as well as locomotion tasks, show that PAWS achieves higher task-specific performance over existing PbRL approaches, highlighting the effectiveness of our method in aligning policies with human preferences.

## 1 INTRODUCTION

Training policies that act according to human intentions is a common goal in machine learning research. For example, autonomous robots must act in accordance with human expectations and desires to increase their acceptance in society. Various concepts have been proposed to obtain those policies, among which imitation learning (IL) (Osa et al., 2018) and reinforcement learning (RL) (Sutton et al., 2018) are the prominent ones. However, both learning paradigms suffer from certain drawbacks. In IL, the agent's performance is upper-bounded by the available data, such that experts are required to collect high-quality data sets. In contrast, RL achieves high-quality policies, but a time-intensive reward design is required to successfully solve a task. Furthermore, embedding a human-like behavior in this reward function is not straightforward and can result in undesired behavior (Skalse et al., 2022). Instead of designing reward functions, preference learning methods propose optimizing a policy based on direct human feedback in the form of pairwise comparisons, also referred to as preferences. The field of preference learning (Wirth et al., 2017) has been largely dominated by methods that utilize RL (Christiano et al., 2017), and referred to as Preference-based Reinforcement Learning (PbRL). These approaches have also been applied to train large language models, where they are more commonly referred to as reinforcement learning from human feedback (RLHF) (Ouyang et al., 2022a). Recently, direct preference learning (Rafailov et al., 2023) has provided a viable alternative as a reinforcement learning-free method for preference learning.

In the standard preference learning setup (Wirth et al., 2017), an expert is presented with two candidate outputs and provides a label indicating which output is preferred under some implicit criteria. These criteria are often challenging to specify mathematically, and the quality of human feedback can vary substantially. In sequential decision-making problems, such as in robot control, the compared outputs are typically multi-step state, or state-action trajectories $\tau = (s_1, a_1, ...)$. When choosing between trajectories $\tau_1$ and $\tau_2$, the expert provides only a binary label, without finer-grained information about individual state–action pairs. Although the trajectory-level preference assignment is the natural way in which humans judge over different outputs, PbRL methods usually rely on atomic transitions within the trajectory to train a policy based on a reward function,

or another value-based utility function, inferred from preference data. This way of optimizing the policy contradicts the information provided by the inferred reward function that is trained using the trajectory-level preferences provided by humans rather than single state-action pairs. Because of this contradiction, proper reward assignment per state-action data is unclear, posing a challenge in policy optimization, and is referred to as the temporal credit assignment problem (Wirth et al., 2017). Fig. 2 (right) visualizes the problem of breaking down the learned trajectory-level rewards to single per-step rewards. Instead of the reward, we use the advantage function (Baird, 1994), which is the value-based utility function in our method and has the same effect on the optimization. Here, darker shades of green indicate higher advantage, and darkest red has the lowest. The preference loss function acts on the sum over individual state-action pair outputs, such that there exist many possible linear combinations of the per-step advantage values to obtain a target output of the advantage for the whole trajectory. This poses a problem when updating the policy based on single state-action pairs because the overall information is saved in the advantage over the whole trajectory rather than in single state-action pairs. We illustrate this relation in Fig. 1. Here, we train an advantage model with all segments and rank all the data according to the expert advantages coming from a trained SAC policy (Haarnoja et al., 2018). This would be infeasible in typical settings, as it would require $\binom{n}{2}$ pairwise preferences in order to compare all segments. Each entry in the top figure visualizes the advantage over a whole trajectory, where the learned advantage model correctly assigns a high value to data points that are also assigned a high advantage from the expert's model, indicated by the monotonically increasing behavior. Visualizing the advantages for individual state-action samples, we do not observe this monotonic behavior, indicating an ill-posed credit assignment (bottom).

To address these shortcomings, we introduce **Preference Learning with Advantage-weighted Segments (PAWS)**[1]. PAWS explicitly takes into account the trajectory-level preferences by updating the policy using an advantage function over trajectory segments. Moreover, PAWS takes inspiration from previous trust-region constrained policy optimization methods that act on the trajectory-level (Kober & Peters, 2008) and adapts the optimization scheme to segment-wise policy updates to maintain the information in the trajectory-based preferences and avoid the single-step reward assignment problem (see Fig. 2 (left)). This policy optimization step leverages the advantage function's understanding of preferences over trajectories and segments of trajectories to update the policy towards preferred trajectory regions, while staying close to the data distribution due to the trust-region constraint. The latter is an important feature that helps avoid updating the policy too greedily based on the advantage function, thereby stabilizing the training.

Our contributions are threefold. First, we analyze the temporal credit assignment problem in preference-based reinforcement learning, clarifying its impact on preference learning. Second, we introduce a novel segment-based preference learning method that explicitly addresses temporal dependencies by utilizing the learned advantage function, thereby enabling more effective propagation of preference signals and mitigating the challenges posed by unclear temporal credit assignment. Finally, we conduct extensive empirical evaluations on a diverse suite of simulated robot manipulation tasks, demonstrating that our approach achieves superior performance over baselines in most tasks.

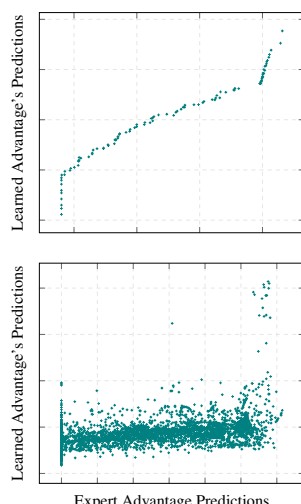

Figure 1: Advantage values for whole trajectories are correctly predicted, indicated by the monotonically increasing behavior (top), whereas per-step advantages do not show a consistent prediction behavior (bottom).

## 2 RELATED WORK

**Preference-based Reinforcement Learning (PbRL)** leverages comparative feedback as the primary supervision signal, distinguishing it from traditional supervised learning (Wirth et al., 2017). Its recent surge in popularity stems largely from applications in fine-tuning large language models (LLMs) (Ouyang et al., 2022b; Lambert, 2025), though our work extends outside the language

---

[1]Code repository: https://github.com/PAWS-ICLR26/PAWS-ICLR26

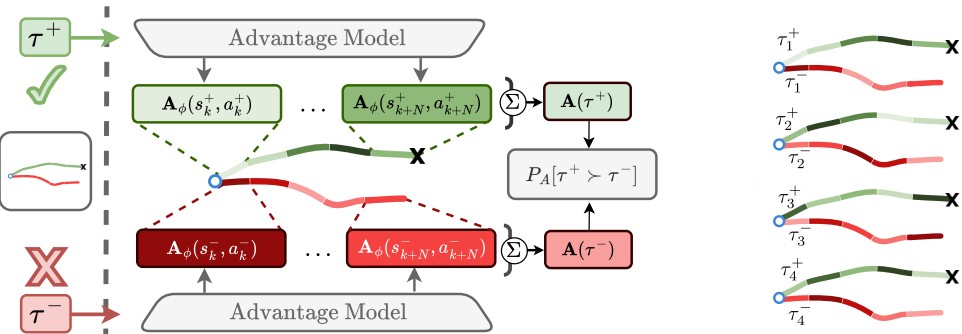

Figure 2: **The Temporal Credit Assignment Problem. Left:** visualizes learning the advantage value $A(\tau)$ from preference data. The advantage model gets positive (green) ($\tau^+$) and negative (red) ($\tau^-$) labeled trajectories as preferences as input (left side) and learns to predict which trajectory data to prefer using the loss function $P_A[\tau^+ \succ \tau^-]$. Notably, this loss is based on the sum of individual state-action pairs $A(\tau^+) = \sum_k A(s_k^+, a_k^+)$, such that the advantage model is free to choose what single advantage values $A(s_k^+, a_k^+)$ to return, as long as the sum over the whole trajectory $A(\tau)$ minimizes the loss function. This unspecified per-step credit assignment is visualized using different shades of the colors in the respective parts of the trajectories, and **right:** shows other opportunities how to potentially assign the per-step advantage values.

domain. While preference-learning pipelines often comprise three conceptual stages, feedback collection, reward or utility-based value function learning, and policy optimization. However, not all methods require or iterate through each component explicitly. Some approaches integrate entropy-based regularization (Lee et al., 2021) or behavior cloning pretraining, where models are initialized using demonstrations of desired behavior. In LLM fine-tuning, a KL-divergence term is frequently introduced to anchor the trained policy to its pretrained counterpart (Ziegler et al., 2019; Stiennon et al., 2020; Ouyang et al., 2022b; Guo et al., 2025). Online preference learning methods such as proposed in Verma & Metcalf (2024), tackle the issue of credit assignment. However, they rely on online preference learning and substantially more data. Gao et al. (2024) propose a similar offline preference learning approach but it depends on massive amounts of unlabeled data.

Direct preference learning methods, such as DPO (Rafailov et al., 2023), CPL (Hejna et al., 2024), IPL (Hejna & Sadigh, 2023), DPPO (An et al., 2023) and PPL (Cho et al., 2025) implicitly avoid the reward assignment problem. Here, the reward function is not learned, and the policy is explicitly optimized using preferences by directly optimizing policy likelihood. However, these methods cannot utilize the reward models nor capture complex dependencies between different state-action pairs. Therefore, they often face challenges in tasks that require complex reasoning (Ivison et al., 2024; Xu et al., 2024). In addition, when preference data is limited, we find these methods often underperform compared to behavior cloning (Table 1).

**(i) Feedback Collection.** The process of acquiring preference data varies significantly across methods. Certain frameworks gather feedback online, collecting trajectories or comparisons dynamically during training (Lee et al., 2021; Rafailov et al., 2023), while others rely on static, pre-collected datasets (Hejna et al., 2024). The generation of preference queries also admits multiple strategies (lee). Our work focuses on offline datasets, though the proposed method remains agnostic to the data collection process and could equally accommodate online settings. A critical consideration in feedback collection is the quality of preference labels (lee). Many approaches address noisy or conflicting data by aggregating signals from multiple policies or annotators. However, our emphasis lies in informative preferences, where an oracle provides labels with access to expert metrics, such as ground-truth rewards or Q-functions. Combining preferences with other feedback modalities, such as demonstrations, is also an open challenge that has been tackled in several recent works (Biyik et al., 2021; Taranović et al., 2023; Ibarz et al., 2018). Other feedback types have also been explored in recent works (Abdolmaleki et al., 2025; Myers et al., 2021; Princeton NLP, 2025).

**(ii) Reward Function Learning.** At this stage, the goal is to train a reward Ng et al. (2000) or a utility-based value function model (Schulman et al., 2015b) that aligns with the collected pref-

erences. The Bradley–Terry (BT) model (Bradley & Terry, 1952) serves as a foundational approach for many methods. In this approach, human preference is modeled as a Boltzmann rational distribution (Baker et al., 2009) over the sum of discounted rewards of individual time steps, which is also referred to as the discounted partial return $P[\tau^+ \succ \tau^-] = \frac{\exp(R(\tau^+))}{\exp(R(\tau^+))+\exp(R(\tau^-))}$. However, recent research suggests that human preferences are better captured through regret-based formulations (Knox et al., 2024). In this approach, the quality of a segment is modeled as a sum of the advantages of the state-action pairs $R(\tau) = \sum_{t=i}^{T} \gamma^t A(s_t, a_t)$. In this paper, we also follow this approach, and refer to $R(\tau)$ as $A(\tau)$. The advantage function $A^\pi(s, a)$ is commonly used in reinforcement learning to measure how much better or worse an action is compared to the expected value of the state under a policy $\pi$. It is defined as the difference between the action-value function $Q^\pi(s, a) = \mathbb{E}_\pi\left[\sum_{t=0}^{\infty} \gamma^t r_t \mid s_0 = s,\ a_0 = a\right]$ and the value function $V^\pi(s) = \mathbb{E}_\pi\left[\sum_{t=0}^{\infty} \gamma^t r_t \mid s_0 = s\right]$, $A^\pi(s, a) = Q^\pi(s, a) - V^\pi(s)$.

Most BT-inspired methods optimize log-likelihood objectives, but alternatives exist, including logistic separation and hinge loss variants (Taranović et al., 2023). To better reflect the complexity of human preference structures, reward models are often modeled with a Transformer-based architecture (Nakano et al., 2021; Gao et al., 2023; Zhao et al., 2024) capable of learning non-Markovian rewards (Kim et al., 2023). When feedback is collected online, the reward model must be continually updated to incorporate new data, adding a layer of dynamic adaptation to the learning process (Taranović et al., 2023; Lee et al., 2021; Cho et al., 2025).

**(iii) Policy Optimization.** Once a reward or value function is learned, policy optimization typically proceeds using RL algorithms. The choice of method often depends on the data collection paradigm. For online settings, trust-region constrained on-policy approaches (Schulman et al., 2015a; Abdolmaleki et al., 2018; Li et al., 2024) or off-policy algorithms such as SAC, where the replay buffer is relabeled with the learned reward, may be employed (Lee et al., 2021). In offline scenarios, algorithms like IQL (Kostrikov et al., 2021) are frequently adopted to handle the constraints of static datasets. In the reward-weighted regression setup (Peters & Schaal, 2007; Neumann & Peters, 2008) and related EM-like approaches (Peters & Schaal, 2007; Abdolmaleki et al., 2018), the constrained policy optimization usually leads to a weighted maximum likelihood optimization that avoids querying the credit assignment out of distribution (Kostrikov et al., 2021; Nair et al., 2020). In the domain of LLM fine-tuning, the learned reward function is commonly paired with policy optimization techniques such as PPO, augmented with KL-regularization to prevent catastrophic divergence from the pretrained model (Schulman et al., 2017; Ouyang et al., 2022b). An alternative and increasingly popular paradigm is the aforementioned Direct Preference Optimization (DPO) (Rafailov et al., 2023; Lambert et al., 2024; Princeton NLP, 2025). However, our method diverges from conventional PbRL methods by directly learning the advantage function, thereby obviating the need for separate reward and value function training. Additionally, we propose updating the policy on segments instead of single state-action pairs, similar to those policy optimization methods already presented in the episode-based RL community (Kober & Peters, 2008; Abdolmaleki et al., 2015; Daniel et al., 2016; Li et al., 2024; 2025). We base the policy update scheme on the EM-like, trust-region constrained optimization problem (Peters et al., 2010) in the offline setting.

## 3 PREFERENCE LEARNING WITH ADVANTAGE-WEIGHTED SEGMENTS

In this section, we describe the novel preference learning method introduced in this paper, PAWS. Using previously collected preferences, we train an advantage function $A_\phi(s_t, a_t)$. Then, using this advantage function, we optimize the policy $\pi(\tau)$ via advantage-weighted segments $A(\tau)$.

**Notation.** In preference learning, we aim to optimize a policy $\pi_\theta : \mathcal{S} \times \mathcal{A} \to \mathcal{R}^+$, where $\mathcal{S}$ and $\mathcal{A}$ denote the respective state and action space. The policy $\pi_\theta$ spans a distribution over the action space and is parameterized by $\theta$. Optimizing the policy $\pi_\theta$, involves optimizing the parameters $\phi$ of an advantage function $A_\phi : \mathcal{S} \times \mathcal{A} \to \mathcal{R}$, that returns a scalar value given state-action pairs. Moreover, we assume having access to state-action trajectories $\tau_T^i = (s_0^i, a_0^i, ... s_T^i, a_T^i)$ of length $T$, but slightly overload the notation and denote a segment of the trajectory with length $N$ starting at time step $k$ as $\tau_{k:N}^i = (s_k^i, a_k^i, ... s_{k+N}^i, a_{k+N}^i)$, as in preference learning we usually deal with segments of trajectories rather than full trajectories. The superscript $i$ indicates a sample following the segment distribution $\tau \sim p_D(\tau)$ that is unknown, but from which we assume to have access to $K$ segment samples. However, whenever the time information is not essential, we overload the subscript and

denote $\tau_i$ as a generic segment sample, omitting to indicate the length or initial timestep to keep the notation uncluttered. Pairs of trajectory segments with $\tau_i^+$ preferred over $\tau_i^-$ are denoted as the dataset $D_{\text{pref}} = \{\tau_i^+, \tau_i^-\}_{i=0}^n$ of size $n$ and we use $\succ$ to denote the preference relation.

## 3.1 ADVANTAGE LEARNING

In our approach, we assume that we are learning the advantage function $A_\phi(s_t, a_t)$ with parameters $\phi$ as proposed in Knox et al. (2024). This aligns better with how humans express preferences than the common approach of using partial returns. Here, the preference likelihood is defined as

$$P_{A_\phi}\left[\tau^+ \succ \tau^-\right] = \frac{\exp\left(\sum_{\tau^+} \gamma^t A_\phi(s_t^+, a_t^+)\right)}{\exp\left(\sum_{\tau^+} \gamma^t A_\phi(s_t^+, a_t^+)\right) + \exp\left(\sum_{\tau^-} \gamma^t A_\phi(s_t^-, a_t^-)\right)} \tag{1}$$

with the goal to maximize the likelihood under the preference data $D_{\text{pref}}$.

Both segments' advantages are computed, and the likelihood optimization reduces to binary cross-entropy optimization of the pairwise difference in cumulative advantage prediction as the sigmoid logit. The loss function then has the form

$$\mathcal{L}_{\text{pref}}(\phi) = -\frac{1}{K} \sum_{D_{\text{pref}}} \log \sigma\left(A_\phi(\tau^+) - A_\phi(\tau^-)\right), \tag{2}$$

where $A_\phi(\tau) = \sum_\tau \gamma^t A_\phi(s_t, a_t)$. This loss function allows using various architectures for $A_\phi(\tau)$, where we restrict ourselves to a transformer-based (Vaswani et al., 2017) and a multilayer perception-based architecture. The analysis of their performance is in Section 4. Practically, the encoder-only transformer model is given each observation and action in the segment $\tau$ and predicts the advantage values $A_\phi(s_k, a_k), ..., A_\phi(s_{k+N}, a_{k+N})$ through an MLP head for each state-action pair. The MLP advantage function takes state $s_t$ and action $a_t$ as input, and outputs $A_\phi(a_t, s_t)$.

To avoid overfitting, we employ early stopping by tracking the accuracy of our trained model using

$$\alpha_{\text{pref}}(A_\phi, D_{\text{pref}}) = \frac{1}{|D_{\text{pref}}|} \sum_{(\tau^+, \tau^-) \in D_{\text{pref}}} \mathbf{1}\left[A_\phi(\tau^+) > A_\phi(\tau^-)\right], \tag{3}$$

where we stop training when $\alpha_{\text{pref}}$ is smaller than a threshold value $\alpha$, or predefined maximum update steps are reached.

## 3.2 POLICY UPDATE

Preference data often provide labels over entire trajectories, or more commonly over segments $\tau$, which is used to train the advantage function $A_\phi(s, a)$ as described in Section 3.1. We use this advantage function to extract a policy $\pi_\theta(a|s)$ such that its likelihood is high for action regions that are assessed as preferable by the advantage function $A_\phi(s, a)$. However, as motivated in Section 1 and in Fig. 2, PAWS aims to update the policy based on the segment data rather than single state-action pairs to avoid the *temporal credit assignment problem*.

Importantly, this policy optimization is based purely on offline data, which requires special treatment to ensure that the policy is not optimized towards action regions that are barely covered by the ground truth data. The latter is specifically problematic when querying the advantage function $A_\phi$ in out-of-distribution action regions because it is unclear how the advantage function extrapolates in these regions. Therefore, it is important to query the advantage function using the data within the data set (Kostrikov et al., 2021; Nair et al., 2020). Because of these reasons, we propose optimizing the policy using a constrained optimization problem in the segment space $\tau$ rather than directly in the action space. Formally, the optimization problem is given as

$$\max_\pi \int \pi(\tau) A_\phi(\tau) d\tau \quad \text{s.t. } \text{KL}\left(\pi(\tau) \| p_D(\tau)\right) \leq \epsilon \text{ and } \int \pi(\tau) d\tau = 1. \tag{4}$$

This optimization problem has two advantages that are crucial in our setting. First, it includes a trust-region that constrains the new policy to be close to the data distribution $p_D$, and second, finding a new policy does not involve sampling new actions from the current policy and querying the

advantage function $A_\phi$ using these samples, as shown in Proposition 1. Both advantages are important to avoid out-of-distribution samples and were previously discussed in the policy fine-tuning literature by Nair et al. (2020). Note that the above optimization problem was already proposed in the reinforcement learning literature by several works (Neumann & Peters, 2008; Peters et al., 2010; Daniel et al., 2016; Abdolmaleki et al., 2018; Nair et al., 2020). Here, we leverage the policy extraction in preference learning using preference data only.

Using Lagrangian optimization (Boyd & Vandenberghe, 2004), we can obtain the optimal solution to the optimization problem in Eq. (4) that we summarize in the following proposition.

**Proposition 1** (Optimal Policy). *(Peters et al., 2010) Given an advantage function $A_\phi(\tau) = \sum_t \gamma^t A_\phi(s_t, a_t)$ and samples from a preference distribution $\tau \sim p_D(\tau)$, the optimal policy to the optimization problem in Eq.* (4) *is given by*

$$\pi^* \propto p_D(\tau) \exp\left(\frac{1}{\lambda} A_\phi(\tau)\right). \tag{5}$$

Notably, the solution shows that the new policy $\pi^*$ is proportional to the preference distribution $p_D$, but weighted with the exponential of the advantage function. The derivation of $\pi^*$ follows Peters et al. (2010) and uses Lagrangian optimization techniques. In Section A, we provide the detailed derivation for the segment-based case as considered here.

Although Proposition 1 gives us an elegant solution to the optimal policy $\pi^*$, it does not provide a parametric form of the policy, and therefore, it is not straightforward to generate samples for new states during inference. Nonetheless, we can fit a parametric policy $\pi_\theta(\tau)$ based on the maximum likelihood objective $\mathcal{L}_{ML}(\theta) = \mathbb{E}_{\pi^*}[\log \pi_\theta(\tau)]$. However, this objective requires samples from the optimal distribution $\pi^*$, making it difficult to optimize. Fortunately, we can still rely on samples from the preference distribution $\tau \sim p_D(\tau)$ that is available offline by leveraging importance sampling

$$\mathcal{L}_{ML}(\theta) = \mathbb{E}_{p_D}[w(\tau) \log \pi_\theta(\tau)], \tag{6}$$

where $w(\tau) = \frac{\pi^*(\tau)}{p_D(\tau)} \propto \exp\left(\frac{1}{\lambda} A_\phi(\tau)\right)$. Where Eq. 6 is derived from a general maximum likelihood objective, i.e., $\mathcal{L}_{ML}(\theta) = \mathbb{E}_{\pi^*}[\log \pi_\theta(\tau)] = \int_\tau \pi^*(\tau)[\log \pi_\theta(\tau)] d\tau = \int_\tau p_D(\tau)\frac{\pi(\tau)}{p_D(\tau)}[\log \pi_\theta(\tau)] d\tau = \mathbb{E}_{p_D}[w(\tau) \log \pi_\theta(\tau)]$.

Similar optimization schemes for obtaining a parametric distribution were suggested, for example, by Peters et al. (2010); Abdolmaleki et al. (2018).

A notable issue in the objective in Eq. (6) is that the evaluation of the likelihood expects whole segments rather than single state-action pairs, which involves the environment dynamics $p(s_{t+1}|s_t, a_t)$ and the state distribution $p(s_0)$, as $\pi_\theta(\tau) = p(s_0) \prod_{t=0}^{T} p(s_{t+1}|s_t, a_t)\pi_\theta(a_t|s_t)$. However, it can be shown that the gradient w.r.t. the parameters $\theta$ does not depend on the environment dynamics and the state distribution, such that the policy update is straightforwardly applicable as shown in the following proposition.

**Proposition 2** (State-Action Gradient). *The gradient of the Objective in Eq.* (6) *w.r.t. the policy parameters $\theta$ is given by*

$$\nabla_\theta \mathcal{L} = \mathbb{E}_{p_D}\left[\exp\left(\frac{1}{\lambda} \sum_t \gamma^t A(s_t, a_t)\right)\left(\sum_t \nabla_\theta \log \pi_\theta(a_t|s_t)\right)\right], \tag{7}$$

*which does not involve the environment's dynamics $p(s_{t+1}|s_t, a_t)$ and the state distribution $p(s_0)$.*

The detailed derivation of this gradient is provided in Section B. Intuitively, the weighting $w(\tau)$ shifts the policy's likelihood to be higher where the advantage of the respective trajectory $\tau$ is high. This can also be seen when calculating the gradient in Eq. 7. Importantly, the Lagrangian multiplier $\lambda$ influences this shift. Very high $\lambda$ values yield near-uniform weights $w(\tau)$, such that the policy has a high likelihood wherever the preference distribution's likelihood is high. For very small $\lambda$ values, the weights are dominated mainly by a small number of segment samples with high advantage values, such that the resulting policy $\pi_\theta$ has high likelihood only for these samples. Finding a good value for $\lambda$ is therefore crucial to balance the exploitation of data points with very high advantage values, especially in rare data cases where the advantage function $A_\phi(\tau)$ might not provide accurate estimates. In the next section, we show how we can obtain the optimal $\lambda^*$ value using the number of effective samples, which is a more intuitive parameter for the user.

### 3.3 OPTIMIZING $\lambda$

Optimizing the parameters $\theta$ of the policy using the loss function in Eq. (6) requires setting a value for $\lambda$. However, $\lambda$ is the Lagrangian multiplier to the trust-region constraint in the optimization problem in Eq. (4) and depends on the parameter $\epsilon$ that upper bounds how much the policy $\pi$ is allowed to deviate from the preference distribution $p_D$. Hence, manually setting $\lambda$ might lead to too conservative or too optimistic policies. Therefore, we follow prior works (Peters et al., 2010; Daniel et al., 2016) and optimize for the Lagrangian multiplier $\lambda$ by minimizing the dual function to the optimization problem in Eq. (4).

**Proposition 3** (Optimal Lagrangian multiplier). *(Peters et al., 2010) Minimizing the dual function*

$$g(\lambda) = \lambda\epsilon + \lambda\log\int p_D(\tau)\exp\left(\frac{A_\phi(\tau)}{\lambda}\right)d\tau \tag{8}$$

*yields the optimal Lagrangian multiplier $\lambda^*$.*

A detailed derivation of the dual function can be found in Section A.1. Importantly, the dual function involves calculating the expectation under the unknown $p_D(\tau)$. However, we can easily use the available samples from $p_D(\tau)$ for a Monte Carlo estimation of the respective expectation.

Yet, finding a good $\epsilon$ might require several iterations of evaluations because the Kullback-Leibler (KL) divergence depends on the dimensionality of the underlying action space. Instead of tuning this hyperparameter, we propose automatically adapting $\epsilon$ based on the desired number of effective samples $n_{\text{eff}}$, which has a more intuitive interpretation. More concretely, we measure $n_{\text{eff}}$ based on the importance weight

$$n_{\text{eff}} = \frac{(\sum_i w_i)^2}{\sum_i w_i^2}, \quad \text{where} \quad w_i = \exp\left(\frac{1}{\lambda}\sum_t \gamma^t A(s_t^i, a_t^i)\right). \tag{9}$$

The number of effective samples is correlated with the hyperparameter $\epsilon$ by the Lagrangian multiplier, and hence, we can define a desired value $n_{\text{eff}}^*$, which indirectly defines the corresponding $\epsilon$ value. Choosing a value for $n_{\text{eff}}^*$ is more intuitive because it defines the number of samples that have a common practical support by both distributions $\pi^*$ and $p_D$. Additionally, adapting $\epsilon$ for a chosen value $n_{\text{eff}}^*$ is straightforward. We can simply optimize the dual function in Eq. (23) to obtain $\lambda^*$, which we then use to calculate $n_{\text{eff}}$ in Eq. (9). Based on this value we can increase $\epsilon$, if $n_{\text{eff}} \gg n_{\text{eff}}^*$, allowing for more greedy updates, or decrease $\epsilon$, if $n_{\text{eff}} < n_{\text{eff}}^*$. We found that $n_{\text{eff}} = 10\%$ leads to good results. A detailed analysis of the effect of choosing $n_{\text{eff}}$ is in Section 4.1.

## 4 EVALUATIONS

We analyze the learning behavior of PAWS using two different parameterizations for the advantage function. Namely, we consider a transformer-based and MLP-based advantage function, which we denote as PAWS (Trans.) and PAWS (MLP), respectively. As baselines, we evaluate Behavior Cloning (BC), P-IQL, CPL (Hejna et al., 2024), CPL+KL (closely related to the PPL method from Cho et al. (2025)), Preference Transformer (Kim et al., 2023), and DPPO (An et al., 2023). We further compare performance under two preference budgets, namely 50 and 500 preferences, sampled from the same dataset. We follow the sparse sampling setup proposed in CPL (Hejna et al., 2024), where we sample from $N$ samples and compare one sample from the first half with a different sample from the second half. Our evaluations are performed on 10 different Metaworld tasks (Yu et al., 2020). However, we modified the task in two ways, as proposed in our baselines. Namely, we randomize the initial hand position and also remove proprioceptive history from the observation. Additionally, we also evaluated on 3 locomotion tasks (Towers et al., 2024), namely HalfCheetah, Hopper, Walker2d. More details about these tasks are in Appendix E.

For each task, we created a dataset that contains equal number samples from 4 policies of varying quality. A comprehensive performance evaluation of these policies is reported in Table 8 in Section E. Generating data in this way represents the different expertise levels of human data more realistically, as suboptimal behavior stems from inexperience and is not the result of optimal behavior with added uncorrelated Gaussian action noise. We use SAC (Haarnoja et al., 2018) to train an

Table 1: Task success (%) $\pm$ 2SE. Best results for $n = 50$ and $n = 500$ are highlighted with orange and teal, respectively. The last row reports the average performance across tasks, showing improvements (in percentage) over standard Behavior Cloning (BC).

| Task | Methods | | | | | | | | | | | | | | | |
|---|---|---|---|---|---|---|---|---|---|---|---|---|---|---|---|---|
| | BC | | P-IQL | | CPL | | CPL+KL | | Pref Trans. | | DPPO | | PAWS (Trans.) | | PAWS (MLP) | |
| *#Preferences* | *50* | *500* | *50* | *500* | *50* | *500* | *50* | *500* | *50* | *500* | *50* | *500* | *50* | *500* | *50* | *500* |
| Button Press | 69±3 | 67±3 | 72±6 | 77±5 | 70±8 | 87±5 | 67±7 | 86±3 | 71±5 | 77±2 | 15±4 | 15±3 | 80±5 | 84±4 | 82±6 | 82±4 |
| Door Open | 48±6 | 52±3 | 52±8 | 83±3 | 36±9 | 71±9 | 44±11 | 77±6 | 62±6 | 87±2 | 15±5 | 15±6 | 70±8 | 96±1 | 65±15 | 98±1 |
| Drawer Open | 54±5 | 60±6 | 58±6 | 71±2 | 53±5 | 79±3 | 56±7 | 78±5 | 45±3 | 71±1 | 9±3 | 13±2 | 44±11 | 74±3 | 40±12 | 75±3 |
| Faucet Close | 51±6 | 66±3 | 59±7 | 80±3 | 53±7 | 64±4 | 48±6 | 63±3 | 59±3 | 85±2 | 22±5 | 32±5 | 67±10 | 87±3 | 68±8 | 87±3 |
| Lever Pull | 36±7 | 44±3 | 31±5 | 38±4 | 28±8 | 46±3 | 29±8 | 47±4 | 34±3 | 47±2 | 6±1 | 4±1 | 28±5 | 58±5 | 30±12 | 55±4 |
| Peg Insert Side | 33±7 | 48±3 | 31±6 | 78±5 | 32±6 | 68±4 | 27±8 | 67±4 | 33±3 | 80±1 | 1±0 | 1±0 | 24±7 | 81±4 | 23±9 | 82±3 |
| Plate Slide | 47±5 | 50±6 | 50±4 | 74±7 | 42±5 | 65±3 | 42±7 | 65±5 | 55±3 | 78±2 | 17±5 | 10±2 | 48±6 | 74±5 | 49±9 | 78±5 |
| Push Back | 25±4 | 37±2 | 23±5 | 43±3 | 25±7 | 45±2 | 18±5 | 43±4 | 25±2 | 48±2 | 4±1 | 3±1 | 26±5 | 56±3 | 24±4 | 53±3 |
| Sweep Into | 30±6 | 58±3 | 35±5 | 66±5 | 26±9 | 66±5 | 31±6 | 64±6 | 31±2 | 66±2 | 9±2 | 9±2 | 35±8 | 74±3 | 36±9 | 74±4 |
| Window Close | 69±8 | 91±2 | 78±9 | 97±1 | 64±7 | 85±5 | 59±8 | 83±4 | 86±0 | 96±1 | 29±7 | 35±3 | 94±5 | 98±1 | 91±5 | 99±0 |
| Avg. over tasks | 46.2 | 57.3 | 48.9 | 70.7 | 42.9 | 67.6 | 42.1 | 67.3 | 50.1 | 73.5 | 12.7 | 13.7 | 51.6 | 78.2 | 50.8 | 78.3 |
| Improvement (%) | 0.0 | 0.0 | 5.8 | 23.4 | -7.1 | 18.0 | -8.9 | 17.5 | 8.4 | 28.3 | -72.5 | -76.1 | 11.7 | 36.5 | 10.0 | 36.6 |

expert policy, saving checkpoints during training. We then automatically selected those checkpoints whose performance during training was such that 4 different qualities, ranging from poor to good, were observed. We generate $10\,000$ rollouts per policy, from which we randomly sample $10\,000$ segments given as consecutive subsets with length $64$. To generate the preference labels, we use the best policy to model the expert nature of the labeler, specifically, the log probabilities of the best policy. As we are using the optimal policy, the policy probabilities follow a Boltzmann distribution with respect to the Q-function. Therefore, it is expected to have same preference labels if used the Q-function directly.

We evaluated PAWS and the baselines on a significantly smaller datasets. To control for sampling variability, all methods are trained on identical datasets for each random seed. This means we initially created the dataset for each seed by sampling from the original dataset to avoid any performance difference that could occur due to dataset variance.

**Metaworld evaluation.** The success rates on Metaworld tasks are reported in Table 1, and the mean returns are provided in Table 5 in Section C. Note that BC values are calculated from the evaluations of CPL, as it performs behavior cloning in its first $200\,000$ steps. We apply the same evaluation principle as in CPL and CPL+KL. At each evaluation point, we run 25 rollouts, average over eight neighboring points and report the maximum among those. All experiments are run over 10 seeds, where we report two times the standard error in addition to the mean. Note that all methods use the same MLP policy network, where we used $n_{\text{eff}} = 10\%$ for PAWS for all evaluations. We use the value of discount factor $\gamma = 1$. For all baselines, we use the code provided by (Hejna et al., 2024) and their proposed hyperparameters (Section F). We can observe that our method, on average, over all tasks achieves higher success rates, as well as mean returns (Table 5). Moreover, in the sparse data case (50 preferences), we do not have a lot of data, the additional signal from the preferences is smaller, and in some tasks, BC is the best-performing method. **Locomotion evaluation.** The average returns from 25 rollouts over 10 seeds are presented in Table 2. The evaluation procedure is the same as for the Metaworld tasks, with the same hyperparameters, with the same network size for policy, and we used $n_{\text{eff}} = 30\%$ for PAWS for all evaluations.

### 4.1 ABLATIONS

**Varying Number of Effective Sample Size.** As discussed in Section 3.3, choosing the number of effective samples $n_{\text{eff}}$ determines how much the updated policy is allowed to deviate from the preference data distribution. A higher value of $n_{\text{eff}}$ yields a higher support between the new policy and the preference distribution, whereas a smaller value pushes the policy towards high advantage action regions. Therefore, for advantage functions that can assess the quality of segments very precisely, one

Table 2: Average episode returns $\pm$ 2SE. Best results for $n = 50$ and $n = 500$ are highlighted with **orange** and **teal**, respectively

| Task | BC | | P-IQL | | CPL | | CPL+KL | | Pref Trans. | | DPPO | | PAWS (Trans.) | | PAWS (MLP) | |
|---|---|---|---|---|---|---|---|---|---|---|---|---|---|---|---|---|
| #Preferences | 50 | 500 | 50 | 500 | 50 | 500 | 50 | 500 | 50 | 500 | 50 | 500 | 50 | 500 | 50 | 500 |
| HalfCheetah | 1019±29 | 1085±66 | 1029±20 | 1031±107 | 998±45 | 950±87 | 998±35 | 948±93 | 945±25 | 968±85 | -90±10 | -97±14 | 1063±26 | 1483±92 | 810±93 | 1630±0 |
| Hopper | 456±45 | 401±22 | 397±39 | 571±65 | 391±39 | 398±35 | 451±61 | 405±32 | 500±28 | 552±48 | 40±59 | 74±31 | 484±41 | 563±60 | 512±57 | 637±50 |
| Walker2d | 167±33 | 266±30 | 116±33 | 947±32 | 119±28 | 446±55 | 123±31 | 476±77 | 150±40 | 726±56 | 35±27 | 32±13 | 208±82 | 997±21 | 235±70 | 1023±16 |

Table 3: Aggregated performance (success rates %) across all tasks. Best results are **highlighted**.

| #Preferences | PAWS (MLP) | PAWS (Transformer) | State (MLP) | State (Transformer) |
|---|---|---|---|---|
| 50 | 51 | **52** | 40 | 44 |
| 500 | **78** | **78** | 67 | 63 |

should choose a small number of effective samples. However, in reality, there are usually not enough preferences available to cover the whole state action space precisely. We therefore study the effect of this parameter on different situations. In Fig. 3b, we consider the case when more data is available (500 preferences) and report the aggregated success rates over the *Peg Insert Side, Sweep Into* and *Drawer Open* for six different relative values for $n_{\text{eff}}$, namely ranging from $5\%$ to $50\%$ of the total number of samples. We use the transformer-based advantage function. Clearly, the performance is higher for a small relative number of effective samples, where at $10\%$ the performance peaks. On the other hand, Fig. 3c reports the performance for the same relative number of effective sample sizes on the *Lever Pull* task, but for only 50 preferences, i.e. a significantly smaller number of preferences. The results indicate that a rather high value of $40\%$ for $n_{\text{eff}}$ yields the best performance of around $45\%$ success rate, outperforming the performance achieved by the BC policy of $36\%$ (see Table 1). This suggests that, given the available data for this specific task, it is challenging to improve upon the BC policy if only a small portion of the available data is effectively used, necessitating that the policy remains closer to the preference data distribution and reduces the greedy updates incentivized by the learned advantage function. Consequently, we suggest choosing $n_{\text{eff}}$ based on each individual problem to obtain the maximal performance. In this work, however, we decided to use a fixed set of hyperparameters to ease the evaluation and found that these hyperparameters already work well, although the performances reported in Table 1 could be improved for PAWS. Note that the results in Fig. 3b and Fig. 3c are plotted over five seeds and may therefore differ from those in Table 1.

**Segments and State-action based updates.** In addition to the results presented in Table 1, we also evaluate policy updates on state-actions using the advantage function to directly compare the benefits of updates on segments. Moreover, we learn an advantage function parameterized either by an MLP or a transformer. We present the aggregated results of the four settings in Table 3. As expected, PAWS has superior performance to the policy updated on states. Additional results for individual tasks are presented in Table 6 and Table 7 in Section D.

**Smaller Segments Size.** We evaluate our method using smaller segments and account for both state updates and segment updates. When a segment consists of only a single step, credit assignment is not an issue. However, this setting is practically infeasible because segments that are too short do not provide enough information to assess their quality. Overall, we consider $1\,000$ segments, and from those we randomly sample $10\,000$ preferences. To ensure a fair comparison and equal numbers of states, we sample additional preferences in proportion to the decrease in segment length. For example, we sample $20\,000$ preferences for a segment of length 32. Fig. 3a reports results are over 5 random seeds. When we perform state updates, the expected performance drop is observed. However, when applying the approach proposed in this work, the performance drop is mitigated by the segment updates. Thus, indicating the benefits of using segment-wise policy updates.

**Spearman's rank correlation coefficient for Segment and State-action based updates.** To evaluate temporal credit assignment, we compute Spearman's rank correlation coefficients. Specifically, we calculate the policy likelihood for each state using three policies: the expert SAC policy, the trained policy using our segment-based updates, and the policy trained with single-step updates. For each segment, we then compute the coefficient between the expert policy likelihoods and those from segment trained policy, as well as between the expert likelihoods and those from single-step updated policy. We evaluate all tasks using five random seeds and 50 preference queries. Overall, across all

Table 4: Task success (%) $\pm$ 2SE with **human-collected preferences**.

| Task | BC | P-IQL | CPL | CPL+KL | Pref Trans. | DPPO | PAWS (Trans.) | PAWS (MLP) |
|---|---|---|---|---|---|---|---|---|
| Button Press | 46±1 | 56±1 | 67±4 | 60±9 | 58±3 | 20±8 | 73±4 | **85±1** |
| Door Open | 69±2 | 69±1 | 50±9 | 66±4 | 79±4 | 7±5 | **91±1** | 89±1 |

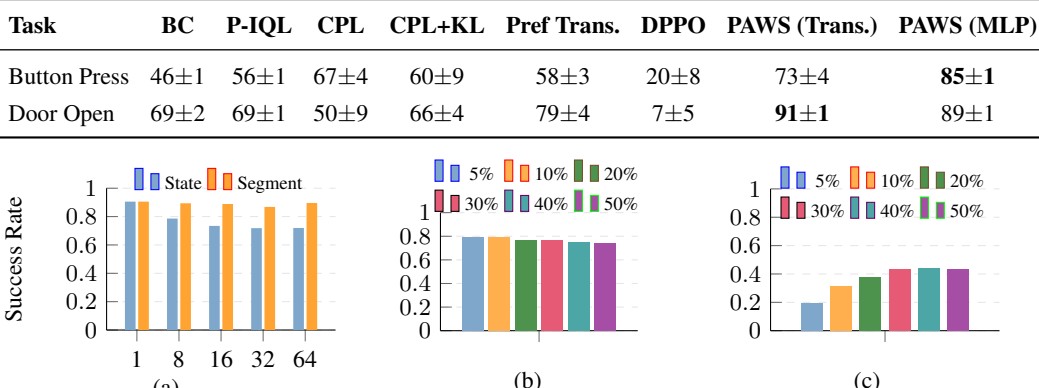

Figure 3: **Ablations on the (a) Segment Length and the (b)-(c) Number of Effective Samples.**
**(a)** Increasing segment length amplifies the *temporal credit assignment problem*, leading to worse
performance when updating the policy using state-action pairs only. This problem is absent when
using segment-based data for the policy update, leading to a consistent performance. Experiments
are done on 5 seeds, and the same number of state-action pairs is used for all segment lengths. **(b-c)**
Effect of Number of Effective Samples for **(b)** 500 Preferences aggregated over the *Peg Insert Side*,
*Sweep Into*, and *Drawer Open* tasks, each with 5 seeds. The results suggest that for a high number
of available data points, a smaller number of effective samples leads to improved results, allowing
the policy to deviate more from the data distribution's support. For a small number of preferences
(50) **(c)** on the *Lever Pull task*, the results suggest using a high relative number of effective samples
to stay close to the data distribution.

tasks, we observe that the mean Spearman's rank correlation coefficient is higher for PAWS. This
indicates that our segment-based approach better preserves the temporal ordering compared to the
step-based approach. Results for each task are presented in the Appendix G.

**Human-label data** Humans can provide different preferences from oracles (Lee et al., 2021). There-
fore, for 2 Metaworld tasks, button press and door open, we collected preferences from a human
labeler. The exact view of the Graphical User interface used to collect the preferences is provided in
the Appendix H. PAWS and all the baselines used the same hyperparameters as in the main evalua-
tion (Table 1). The mean success rates over 5 seeds are presented in Table 4. Overall, the results are
inline with the oracle provided evaluations and indicate the benefits of using PAWS.

## 5 CONCLUSION

We have proposed PAWS, a novel method for preference learning that first learns an advantage func-
tion and subsequently updates the policy exploiting this advantage function purely on offline data.
Notably, PAWS suggests updating the policy based on segment data rather than single state-action
pairs to mitigate the *temporal credit assignment problem* that occurs because the advantage function
is learned on whole segments rather than single state-action pairs and might lead to undesired policy
updates. PAWS takes inspiration from prior works in reinforcement learning that propose a trust re-
gion constrained policy optimization that avoids querying the advantage function out of distribution
by updating the policy only using the available data set. Moreover, PAWS benefits from updating
the policy based on the segments, thereby avoiding the *temporal credit assignment problem*. These
features lead to favorable performance over baselines on 10 different tasks for a varying number of
available preferences. **Limitation and Future Work.** A limitation of PAWS is that the number of
effective samples needs to be tuned depending on the quality of the data set. Additionally, we plan
to extend PAWS to the offline-to-online case, where the advantage and policy update also take into
consideration the newly generated preference data for effective updates. This case is particularly
relevant, as it can be used for fine-tuning models.

## ETHICS STATEMENT

This work follows the ICLR Code of Ethics. It does not involve human subjects, private data, or applications with possible harmful impact. All experiments use public datasets, and we do not see any ethical issues with the data, methods, or results.

## REPRODUCIBILITY STATEMENT

All main experiments are run with 10 seeds for statistical robustness, and results are reported using average values and twice the standard error. Additionally, we include all details needed to reproduce our method in Section 4, and describe the experimental setup in Section F. We will release the full code and datasets with the camera-ready version.

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

## USE OF LLMS

We used Large Language Models (LLMs) as assistive tools in preparing this paper. They were mainly used for grammar correction, rephrasing for clarity, and small suggestions to improve the flow of the text. In addition, we used LLMs to help find related work papers, but all suggested references were carefully checked by us for relevance and correctness. All LLM–generated text was reviewed and edited by the authors to ensure it matches our intentions and is scientifically accurate.

## A  PROOF OF THE OPTIMAL DISTRIBUTION

*Proof.* The Lagrangian to the optimization problem in Eq. (4)

$$L(\pi, \lambda, \beta) = \int \pi(\tau) A_\phi(\tau) d\tau + \lambda \left( \epsilon - \int \pi(\tau) \left( \log \pi(\tau) - \log p_D(\tau) \right) d\tau \right) \tag{10}$$

$$+ \beta \left( 1 - \int \pi(\tau) d\tau \right) \tag{11}$$

$$= \int \pi(\tau) \left( A_\phi(\tau) - \lambda \left( \log \pi(\tau) - \log p_D(\tau) \right) - \beta \right) d\tau + \lambda \epsilon + \beta \tag{12}$$

Obtaining the optimal solution

$$\nabla_\pi L = A_\phi(\tau) - \lambda \log \pi(\tau) + \lambda \log p_D(\tau) - \lambda - \beta = 0 \tag{13}$$

$$\rightarrow \pi^* = \exp \left( \frac{A_\phi(\tau) + \lambda \log p_D(\tau)}{\lambda} \right) \exp \left( \frac{-\beta - \lambda}{\lambda} \right) \tag{14}$$

$$\square$$

### A.1  PROOF OF THE DUAL FUNCTION

*Proof.* We can eliminate the Lagrangian multiplier $\beta$ to the normalization constraint

$$\int \pi(\tau) d\tau = 1 \tag{15}$$

in the optimization problem Eq. (4) by inserting the optimal solution in Eq. (13) into the Eq. (15), such that we obtain

$$1 = \exp \left( \frac{-\beta - \lambda}{\lambda} \right) \int \exp \left( \frac{A_\phi(\tau) + \lambda \log p_D(\tau)}{\lambda} \right) d\tau \tag{16}$$

$$0 = \frac{-\beta - \lambda}{\lambda} + \log \int \exp \left( \frac{A_\phi(\tau) + \lambda \log p_D(\tau)}{\lambda} \right) d\tau \tag{17}$$

$$\beta = -\lambda + \lambda \log \int \exp \left( \frac{A_\phi(\tau) + \lambda \log p_D(\tau)}{\lambda} \right) d\tau \tag{18}$$

By inserting the optimal solution $\pi^*$ in Eq. (14) into the Lagrangian in Eq. (12), we obtain

$$g(\lambda) = \int \pi(\tau) \left( A_\phi(\tau) - A_\phi(\tau) - \lambda \log p_D(\tau) + \beta + \lambda + \lambda \log p_D(\tau) \right) - \beta) d\tau + \lambda \epsilon + \beta \tag{19}$$

$$= \int \pi(\tau) \lambda \tau + \lambda \epsilon + \beta \tag{20}$$

$$= \lambda + \lambda \epsilon + \beta \tag{21}$$

$$= \lambda \epsilon + \lambda \log \int \exp \left( \frac{A_\phi(\tau) + \lambda \log p_D(\tau)}{\lambda} \right) d\tau \tag{22}$$

$$= \lambda \epsilon + \lambda \log \int p_D(\tau) \exp \left( \frac{A_\phi(\tau)}{\lambda} \right) d\tau \tag{23}$$

$$\square$$

## B   DERIVATION OF THE GRADIENT

*Proof.* We can rewrite the more common state action policy optimization pendant to the maximum likelihood objective from Eq. (6) as

$$\mathcal{L}_{ML}(\theta) = \mathbb{E}_{p_D}\left[\exp\left(\frac{1}{\lambda}\sum_t \gamma^t A_\phi(s_t, a_t)\right)\left(\log p(s_0) + \sum_t \log \pi_\theta(a_t|s_t) + \log p(s_{t+1}|s_t, a_t)\right)\right],$$
(24)

which includes the unknown initial state distribution $p(s_0)$ and the transition dynamics $p(s_{t+1}|s_t, a_t)$. However, note that neither of these entities affects the optimization, as they do not depend on the parameter $\theta$ and vanish when calculating the gradient using

$$\nabla_\theta \mathcal{L} = \mathbb{E}_{p_D}\left[\exp\left(\frac{1}{\lambda}\sum_t \gamma^t A_\phi(s_t, a_t)\right)\left(\sum_t \nabla_\theta \log \pi_\theta(a_t|s_t)\right)\right],$$
(25)

which is closely related to Kober & Peters (2008). □

## C   AVERAGE RETURNS

In Table 5 we present the results of mean episode returns.

Table 5: Mean episode returns across methods and preference counts. Best results for $n = 50$ and $n = 500$ are highlighted with orange and teal, respectively.

| Task | P-IQL | | CPL | | CPL+KL | | PAWS (Trans.) | | PAWS (MLP) | |
|---|---|---|---|---|---|---|---|---|---|---|
| | **50** | **500** | **50** | **500** | **50** | **500** | **50** | **500** | **50** | **500** |
| Button Press | 1423±39 | 1593±43 | 1443±86 | 1598±55 | 1416±74 | 1581±29 | 1356±74 | 1568±25 | 1583±81 | 1722±22 |
| Door Open | 870±95 | 1593±43 | 970±126 | 1598±55 | 762±125 | 1581±29 | 822±127 | 1568±25 | 1294±158 | 1819±27 |
| Drawer Open | 1508±80 | 1562±22 | 1463±78 | 1626±44 | 1472±78 | 1646±41 | 1481±61 | 1617+35 | 1104±194 | 1634±47 |
| Faucet Close | 1516±59 | 1692±31 | 1599±79 | 1900±27 | 1516±86 | 1662+61 | 1449±68 | 1661±64 | 1696±133 | 1997±37 |
| Lever Pull | 366±45 | 426±18 | 343±33 | 406±32 | 347±32 | 450±51 | 338±33 | 467±45 | 350±57 | 462±39 |
| Peg Insert Side | 882±87 | 1281±28 | 670±107 | 1511±49 | 836±98 | 1426±74 | 814±125 | 1463±65 | 505±136 | 1545±47 |
| Plate Slide | 1049±89 | 1156±97 | 1095±78 | 1582±78 | 950±124 | 1395±108 | 955±104 | 1344±91 | 1070±130 | 1548±91 |
| Push Back | 312±56 | 484±39 | 270±65 | 607±111 | 261±88 | 629±45 | 203±68 | 584±55 | 363±73 | 872±70 |
| Sweep Into | 410±90 | 1001±57 | 475±85 | 1117±103 | 325±123 | 1142±100 | 439±111 | 1128±88 | 463±99 | 1326 ±85 |
| Window Close | 1095±97 | 1371±34 | 1198±104 | 1530±34 | 1015+106 | 1371±22 | 982+123 | 1358±47 | 1373±90 | 1572±17 |

# D  COMPARISON OF SEGMENT AND STATE-BASED UPDATES

In Table 6 and Table 7, we have the reported the comparison between Segment and State-based updates.

Table 6: Performance results for individual tasks with **50** preferences. Values represent success rates (%) $\pm$ 2SE. Best results are highlighted with **bold**.

| Task | PAWS (MLP) | PAWS (Transformer) | State (MLP) | State (Transformer) |
|------|-----------|--------------------|-------------|---------------------|
| Button press | **82±6** | 80±5 | 72±6 | 74±5 |
| Door open | 65±14 | **70±8** | 40±12 | 42±9 |
| Drawer open | 39±12 | **44±11** | 22±10 | 31±13 |
| Faucet close | **68±8** | 67±10 | 64±9 | 60±8 |
| Lever pull | 30±12 | 28±6 | 16±11 | **32±9** |
| Peg insert side | 23±9 | **24±7** | 19±9 | 20±9 |
| Plate slide | **49±9** | 48±6 | 37±6 | 45±8 |
| Push back | 24±4 | **26±5** | 17±3 | 19±5 |
| Sweep into | **36±9** | 34±8 | 33±6 | 34±5 |
| Window close | 91±7 | **94±5** | 79±8 | 80±8 |

Table 7: Performance results for individual tasks with **500** preferences. Values represent success rates (%) $\pm$ 2SE. Best results are highlighted with **bold**.

| Task | PAWS (MLP) | PAWS (Transformer) | State (MLP) | State (Transformer) |
|------|-----------|--------------------|-------------|---------------------|
| Button press | 82±4 | **84±4** | 75±4 | 78±5 |
| Door open | **98±1** | 96±1 | 82±4 | 64±8 |
| Drawer open | **75±3** | 74±3 | 59±4 | 66±3 |
| Faucet close | **87±3** | **87±3** | 71±10 | 73±5 |
| Lever pull | 55±4 | **58±5** | 56±4 | 55±3 |
| Peg insert side | **82±3** | 81±4 | 68±5 | 55±4 |
| Plate slide | **78±5** | 74±5 | 54±6 | 52±7 |
| Push back | 53±3 | **56±3** | 44±4 | 36±6 |
| Sweep into | **74±4** | **74±3** | 67±3 | 63±2 |
| Window close | **99±0** | 98±1 | 91±2 | 91±3 |

# E    PREFERENCE DATASET GENERATION

We evaluate on MetaWorld environments with the modifications of CPL(Hejna et al., 2024). In contrast to the original MetaWorld tasks (Yu et al., 2020), these are modified by randomizing the goal but including the target in the state observation, as well as randomizing the initial robot position and removing the proprioceptive state history from the observation. Additionally, we evaluate our method on 3 locomotion tasks, namely HalfCheetah, Hopper, Walker2d. For all tasks, we train one policy with SAC(Haarnoja et al., 2018) and choose rollouts from four checkpoints throughout training. For the metaworld tasks, exact step counts, average return, and success rate are given in Table 8.

Table 8: Dataset quality for different tasks

| Task | Policy 1 | | | Policy 2 | | | Policy 3 | | | Best Policy | | |
|---|---|---|---|---|---|---|---|---|---|---|---|---|
| | Step | Return | Success | Step | Return | Success | Step | Return | Success | Step | Return | Success |
| Button Press | 40k | 509 | 15.9% | 70k | 1335 | 72.7% | 150k | 1320 | 73.8% | 240k | 1709 | 95.9% |
| Door Open | 30k | 462 | 2% | 50k | 708 | 41% | 70k | 1314 | 83% | 830k | 1994 | 100% |
| Drawer Open | 210k | 1279 | 8.08% | 270k | 1535 | 61.0% | 290k | 1640 | 76.3% | 350k | 1786 | 91.3% |
| Faucet Close | 30k | 849 | 0.93% | 60k | 1346 | 36.1% | 90k | 1451 | 50.8% | 140k | 2124 | 98.1% |
| Lever Pull | 30k | 204 | 0.00% | 190k | 265 | 23.3% | 300k | 382 | 51.5% | 640k | 720 | 80.0% |
| Peg Insert Side | 340k | 735 | 3.59% | 390k | 1201 | 32.5% | 410k | 1507 | 85.8% | 480k | 1760 | 96.7% |
| Plate Slide | 50k | 487 | 9.21% | 60k | 448 | 8.11% | 120k | 1543 | 78.1% | 250k | 2006 | 99.0% |
| Push Back | 280k | 47 | 11.5% | 290k | 102 | 20.1% | 410k | 573 | 57.4% | 600k | 1622 | 97.1% |
| Sweep Into | 50k | 142 | 0.11% | 150k | 490 | 52% | 300k | 1176 | 81% | 910k | 1958 | 97% |
| Window Close | 30k | 240 | 4.52% | 70k | 669 | 47.9% | 80k | 906 | 13.8% | 120k | 1524 | 98.8% |

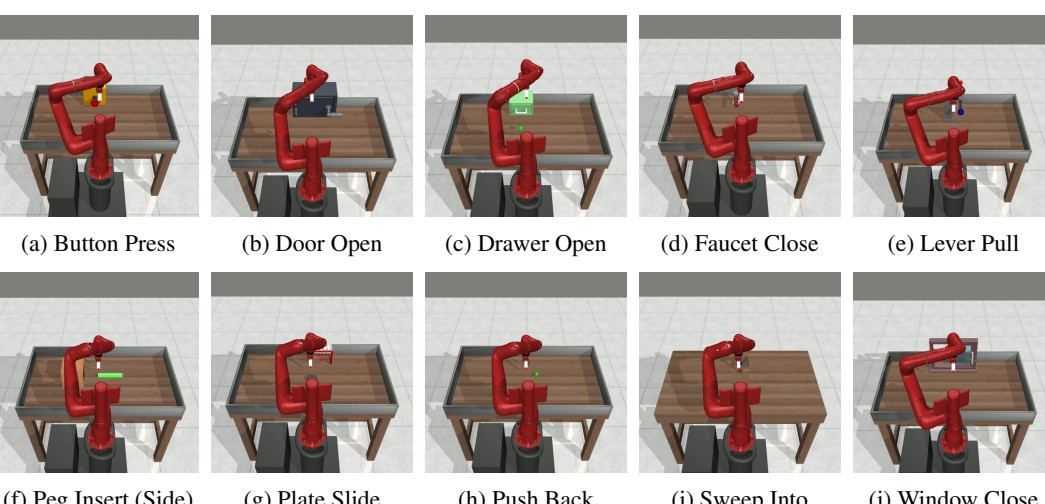

|                      |                   |                    |                     |                    |
|----------------------|-------------------|--------------------|---------------------|--------------------|
| (a) Button Press     | (b) Door Open     | (c) Drawer Open    | (d) Faucet Close    | (e) Lever Pull     |
| (f) Peg Insert (Side)| (g) Plate Slide   | (h) Push Back      | (i) Sweep Into      | (j) Window Close   |

Figure 4: Meta-World manipulation tasks used in our experiments. Each panel shows the initial configuration (a–j).

## E.1    METAWORLD TASK DESCRIPTIONS

We evaluate on ten Meta-World manipulation tasks shown in Fig. 4: *(a) Button Press* press a small button on the panel until it activates; *(b) Door Open* open a hinged door by pulling the handle to a target angle; *(c) Drawer Open* pull the drawer along its rail until it reaches the goal extension; *(d) Faucet Close* rotate the faucet handle clockwise until it is fully off; *(e) Lever Pull* pull the short

lever down through a quarter turn; *(f) Peg Insert (Side)* insert a cylindrical peg into a horizontal side hole without jamming; *(g) Plate Slide* push the plate across the table into the marked goal region; *(h) Push Back* push the movable puck backward to the target position; *(i) Sweep Into* sweep a small object across the surface into a container opening; *(j) Window Close* push the sliding window pane along its track until fully closed.

## E.2 LOCOMOTION TASK DESCRIPTIONS

We evaluate 3 different locomotion tasks (Towers et al., 2024), namely HalfCheetah, Hopper, and Walker2d. All tasks have same episode length of 250 steps, and episodes are finished when the final step is reached. There are no other terminal conditions that are otherwise present in Towers et al. (2024) because this is required for consistent generation of trajectories used for preferences. The tasks are visualized in Fig. 5

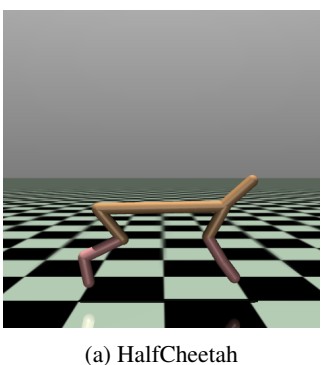 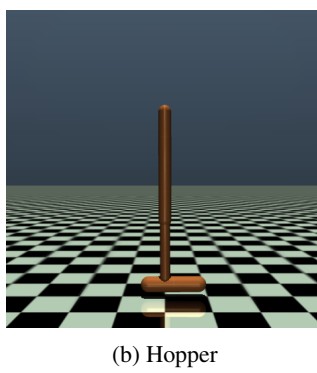 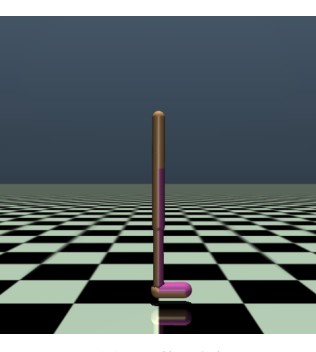

|          (a) HalfCheetah          |          (b) Hopper          |          (c) Walker2d          |

Figure 5: Locomotion tasks.

## F ALGORITHM DETAILS AND HYPERPARAMETERS

In the following tables are listed the method hyperparameters. For baselines in Table 9 and for our method in Table 10. Our method is also shown in Algorithm 1.

Table 9: Hyper-parameters for CPL and variants.

| Hyperparameter | CPL | CPL (KL) | P-IQL | Pref. Transformer |
|---|---|---|---|---|
| Learning Rate | 0.0001 | 0.0001 | 0.0003 | 0.0003 |
| Temp $\alpha$ | 0.1 | 0.1 | - | - |
| Bias $\lambda$ | 0.5 | 0.5 | - | - |
| $\gamma$ | - | - | 0.99 | 0.99 |
| Expectile $\tau$ | - | - | 0.7 | 0.7 |
| Temperature | - | - | 0.3333 | 3.0 |
| Hidden Dims | - | - | - | (256, 256) |
| Soft Target $\tau$ | - | - | - | 0.005 |
| Reward Net Steps | - | - | 50000 | - |
| Evaluation Step | 5000 | 5000 | 5000 | 5000 |

Table 10: Hyper-parameters for PAWS.

| Hyperparameter | PAWS (MLP) | PAWS (Transformer) |
|---|---|---|
| **Actor networks** | | |
| Learning rate | 0.0003 | 0.0003 |
| Dropout | 0.25 | 0.25 |
| Hidden dimension | 512 | 512 |
| Hidden depth | 2 | 2 |
| Evaluation Step | 250 | 250 |
| **Reward networks** | | |
| Learning rate | 0.0003 | 0.0003 |
| Max Len | - | 64 |
| Dropout | - | 0.1 |
| Hidden dimension | 512 | 512 |
| Number of heads | - | 8 |
| Number of layers | 3 | 4 |
| Position encoding | - | learned |
| Min. early stopping value $\alpha$ | 0.995 | 0.995 |
| $\gamma$ | 1.0 | 1.0 |

---

**Algorithm 1 PAWS: Preference Learning with Segment-Level Advantage Optimization**

---

1: **Input:** Offline segment dataset $\mathcal{D} = \{\tau_i\}_{i=1}^K$ sampled from $p_\mathcal{D}(\tau)$
2: **Input:** Offline preference dataset $\mathcal{D}_{\text{pref}} = \{(\tau_i^+, \tau_i^-)\}_{i=1}^n$, where $\tau_i^+ \succ \tau_i^-$
3: **Input:** Advantage model $A_\phi(s, a)$, policy $\pi_\theta(a \mid s)$
4: Initialize $\phi$ (advantage) and $\theta$ (policy)
5: **while** policy $\theta$ not converged **do**

    **1. Advantage learning on preference data**
6:    Minimize preference loss (Eq. (2)):

$$\mathcal{L}_{\text{pref}}(\phi) = -\frac{1}{n} \sum_{(\tau^+, \tau^-) \in \mathcal{D}_{\text{pref}}} \log \sigma(A_\phi(\tau^+) - A_\phi(\tau^-))$$

7:    Update $\phi \leftarrow \phi - \alpha_\phi \nabla_\phi \mathcal{L}_{\text{pref}}$

    **2. Compute importance weights (Eq. (6))**
8:    **for** each $\tau_i \in \mathcal{D}$ **do**
9:        $w_i \leftarrow \exp\left(\frac{1}{\lambda} A_\phi(\tau_i)\right)$
10:    **end for**

    **3. Policy update using weighted maximum-likelihood (Eq. (7))**
11:    Compute gradient:

$$\nabla_\theta \mathcal{L} = \mathbb{E}_{\tau \sim p_\mathcal{D}} \left[ w(\tau) \sum_t \nabla_\theta \log \pi_\theta(a_t \mid s_t) \right]$$

12:    Update $\theta \leftarrow \theta + \alpha_\theta \nabla_\theta \mathcal{L}$
13: **end while**
14: **return** $\pi_\theta$

---

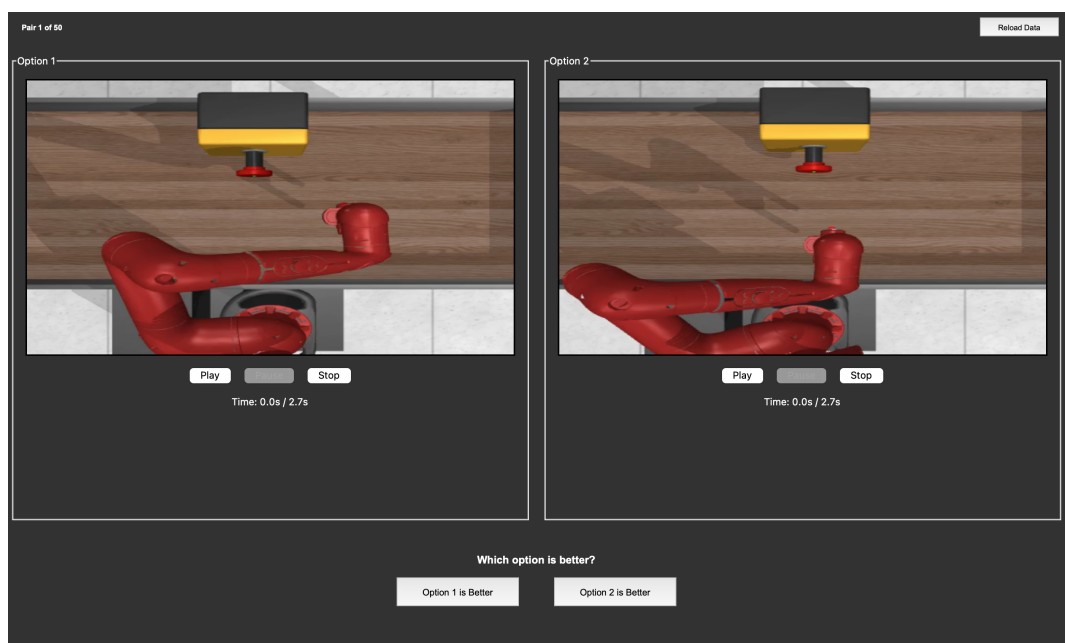

Figure 6: GUI for collecting preference labels from humans.

# G  ABLATION SPEARMAN'S RANK CORRELATION COEFFICIENTS.

In order to compare sequences, we compute the Spearman's correlation coefficient

$$
r_s = \frac{\sum_{i=1}^{n}(\mathrm{rg}_{X_i} - \overline{\mathrm{rg}_X})(\mathrm{rg}_{Y_i} - \overline{\mathrm{rg}_Y})}{\sqrt{\sum_{i=1}^{n}(\mathrm{rg}_{X_i} - \overline{\mathrm{rg}_X})^2}\sqrt{\sum_{i=1}^{n}(\mathrm{rg}_{Y_i} - \overline{\mathrm{rg}_Y})^2}},
\tag{26}
$$

where $\mathrm{rg}_X$ and $\mathrm{rg}_Y$ are the ranked elements of the two vectors that are being compared, and $\overline{\mathrm{rg}_X}$ and $\overline{\mathrm{rg}_Y}$ there respective means. In our evaluation, in one vector are the log likelihoods of the expert policy of a segment and in the other vector the log likelihoods coming from our trained policy for the same segment. We use the transformer-version of the advantage function, and we use 100 segments for each task. The values are means over 5 seeds.

Table 11: Spearman correlation coefficient between likelihoods policy and expert policy for each MetaWorld task. We compare the **State** versus the **Segment** representation.

|  | Button Press | Door Open | Drawer Open | Faucet Close | Lever Pull | Peg Insert Side | Plate Slide | Push Back | Sweep Into | Window Close |
|---|---|---|---|---|---|---|---|---|---|---|
| **State** | 0.164 | 0.276 | 0.162 | -0.153 | 0.030 | 0.089 | 0.016 | 0.036 | 0.124 | -0.018 |
| **Segment** | 0.177 | 0.303 | 0.175 | -0.132 | 0.032 | 0.093 | 0.023 | 0.056 | 0.125 | 0.005 |

# H  HUMAN PREFERENCES

We collected 50 preferences from 100 segments. The preference pairs were generated in the same way as in Section E, with the labels provided by a human. To collect the labels, we use the GUI shown in Fig. 6.

