# OpenReview forum: "PAWS: Preference Learning with Advantage-Weighted Segments"
_ICLR.cc/2026/Conference — Submitted to ICLR 2026_

### Official Review · Reviewer_MXRB · 2025-10-20

**Soundness:** 2
**Presentation:** 3
**Contribution:** 2
**Rating:** 2
**Confidence:** 4

**Summary:**

The paper proposes to use learning an advantage function from preference data to address the credit assignment problem faced when learning a reward model from preference data. The proposed solution is a method named PAWS that aims to preserve segment-level preference information instead of inferring values for single state-action pairs when provided segment level preference information.  The method is applied to the offline data case. The advantage function is learned using the Bradley-Terry objective, and the policy using a combination of standard approaches to learning from offline data. Results are presented for ten tasks from MetaWorld for two different amounts of preference data (50 vs. 500 samples) and for two different advantage function architectures (transformer vs. MLP). Across tasks there is no clear winner between the transformer and MLP architecture. When using 50 preference samples, PAWS performs best on 6/10 tasks and on 8/10 tasks when using 500 samples.

**Strengths:**

- The paper is well written and easy to follow.
- The paper aims to address an important problem in PbRL, which is accounting for the credit assignment problem when learning from preference feedback.
- Multiple baselines are compared against, all of which operate in the same offline setting as PAWS.
- The results demonstrate that PAWS improves performance relative to baselines on 6/10 MetaWorld tasks give 50 preference samples and 8/10 tasks given 500 samples. When PAWS performs best, the gains are not small suggesting the approach improves performance more often than not.

**Weaknesses:**

- No mention of any prior work in PbRL specifically aimed at addressing the credit assignment problem. Please update the related work section to mention prior work has attempted to tackle this problem. Some papers focusing on the credit assignment problem in reward learning include:
     - Zou, Haosheng, et al. "Reward shaping via meta-learning." arXiv preprint arXiv:1901.09330 (2019).
     - Novoseller, Ellen, et al. "Dueling posterior sampling for preference-based reinforcement learning." Conference on Uncertainty in Artificial Intelligence. PMLR, 2020.
     - Gangwani, Tanmay, Yuan Zhou, and Jian Peng. "Learning guidance rewards with trajectory-space smoothing." Advances in neural information processing systems 33 (2020): 822-832.
     - Verma, Mudit, and Katherine Metcalf. "Hindsight PRIORs for Reward Learning from Human Preferences." The Twelfth International Conference on Learning Representations (2024).
- The exact connection between the advantage function and the credit assignment problem needs to be better formulated and motivated in the paper. For example, why/how does advantage address the credit assignment problem?
- The approach to learning the advantage function in Section 3.1 is exactly the approach and method used to learn reward functions. It is not clearly spelled out how the resulting function is an advantage function, and not a reward function used like an advantage function. This calls into question the validity of applying the preference learned model as an advantage. Therefore, how exactly the model learned from the BT objective is an advantage function instead of a reward function needs to be addressed.
- The paper uses many existing optimizations and approaches, but applied to the problem of PbRL. Without a clear motivation for how advantage specifically addresses the credit assignment problem, the contributions of the paper are minimal.
- Results are only presented on MetaWorld. It is common for PbRL papers to present results on both DMC tasks and MetaWorld to demonstrate the algorithm works on goal and non-goal-based tasks. Therefore, to be inline with prior work, results should be presented on non-goal-based tasks.
- The paper does not present results suggesting that the credit assignment problem specifically is addressed. Please include experiments highlighting
- The use of BC and P-IQL as baselines for PAWS is not clear as neither method relies on learning from preference data. Please motivate why these are appropriate baselines.
- Small issues:
     - In Table 2 both PAWS(MLP) and PAWS(Transformer) should have their 500 results bolded. For 50 preferences, is the differences of 51 versus 52 meaningful?

**Questions:**

- For Figure 3 (b) and (c), why are different tasks shown for the 500 versus 50 preference samples cases? Why aggregate over 3 tasks for 500 samples and only present results for one task for 50 samples?

---

> ### Author Response · Authors · 2025-12-03
>
> We thank the reviewer for their detailed and helpful feedback, and especially for pointing out the need for further motivation of using advantage-based preferences. We address the individual concerns below.
>
> W1: We agree with the reviewer that a more comprehensive comparison to existing methods is useful. We added a discussion on the suggested and similar recent approaches to the related work section, and made sure to clarify how they relate to PAWS. Here would like to mention that method are comparable to our work, as they do not
>
> W2:  The main claim of our paper is that policy updates on segments mitigate the negative effects on TCA in preference learning. We support these claims with our ablations with smaller segment size and additonal abblation where we calculated the Spearmans’ rank correlation coeffiect.
>
> W3: Recent work [1] show that human preference are more aligned with an advantage function, rather than the commonly used reward function. We make this point clearer in the paper.
>
> [1]Knox, W. B., Hatgis-Kessell, S., Booth, S., Niekum, S., Stone, P., & Allievi, A. (2023). *Models of human preference for learning reward functions*. Transactions on Machine Learning Research.
>
> W4: We thank the reviewer for raising this issue, and agree with the need to better motivate our contribution. PAWS provides strong empirical evidence for using advantage-based preferences on segments to tackle the credit assignment problem. In addition, its novelty lies in the application of well-known techniques from optimization to preference learning within this advantage-based framework, and in showing the benefits that come with these techniques.
>
> W5: We evaluated PAWS on three additional locomotion tasks from the Deepmind control suite, namely HalfCheetah, Hopper and Walker2d. We found similar trends to the goal-based tasks, namely that PAWS shows better or on par performance as existing baselines. We believe that these additional evaluations paint a clearer picture of PAWS’ strengths and weaknesses compared to existing methods.
>
> W6: We added an additional experiment specifically targeted towards showing the advantages of PAWS for temporal credit assignment. We also adjusted the writing to better show the connection of some existing experiments and evaluations to temporal credit assignment.
>
> W7: We use BC to provide a simple Imitation Learning baseline that essentially shows the inherent performance in the demonstrations without preference information. P-IQL additionally uses pairwise preferences to learn a reward via the Bradley-Terry model. We clarified these motivations in the revision. We also added comparisons to DPPO and Preference Transformers to provide a more holistic evaluation of our method.
>
> W8: We thank the reviewer for catching this oversight. Similar to Table 1, we bolded the highest average value across methods, regardless of overlapping confidence intervals. While we agree that this evaluation makes it difficult to exactly distinguish between PAWS (MLP) and PAWS (Transformer), both methods are clearly superior to their state-based alternatives.
>
> ---
>
> Q1: In Figure 3c we evaluated the Lever Pull with 50 preferences, as this is a task where BC is superior to all the preference-based methods. With this ablation, we wanted to show that with the better selection of effective sample size we can outperfom BC.  For computational reasons we have have restricted our self to 3 tasks for evalution in Figure 3b.
>
> We thank the reviewer again for their detailed and constructive review.

---

### Official Review · Reviewer_s17o · 2025-10-23

**Soundness:** 3
**Presentation:** 3
**Contribution:** 3
**Rating:** 4
**Confidence:** 4

**Summary:**

This paper analyzes how per-step reward inference in preference-based reinforcement learning (PbRL) leads to inconsistencies and instability because of the temporal credit assignment problem. To address this, it proposes PAWS (Preference Learning with Advantage-weighted Segments), an offline preference learning framework that learns a segment-level advantage function and performs segment-wise trust-region constrained policy optimization with adaptive scaling for stable updates within the data distribution. Experiments on robot manipulation tasks demonstrate that PAWS effectively mitigates credit assignment errors and outperforms existing PbRL methods.

**Strengths:**

1. The paper provides excellent visualization (Figures 1 and 2) of the temporal credit assignment problem in preference learning. The empirical demonstration that trajectory-level advantages are well-predicted while per-step advantages are inconsistent effectively motivates the segment-based approach.
2. The paper derives the optimal policy through Lagrangian optimization (Proposition 1) and shows that the gradient does not depend on environment dynamics (Proposition 2), providing solid theoretical foundation.

**Weaknesses:**

1. The temporal credit assignment problem in PbRL has been extensively explored in recent works that explicitly model temporal dependencies through transformer-based reward estimator or world models [1,2,3]. While [2] was proposed for online PbRL, the reward learning component could be adapted to offline settings for comparison. The authors should more clearly articulate the fundamental difference between their segment-based policy update approach and these existing methods that address TCA.
2. Several recent and relevant methods are missing, such as IPL [4], DPPO [5], LiRE [6], and APPO [7]. At minimum, comparison with IPL and DPPO (both avoiding reward modeling) would strengthen the empirical contribution.
3. The paper generates preference labels by comparing the log probabilities under the best policy, claiming it “models the expert nature of the labeler”. This design choice is neither deterministic (as in reward-based oracle preferences for fair comparison) nor genuinely human (as in real-world situation), and no prior work is cited that justifies this design choice. The log probability approach is unconventional and its validity is not established.
4. All experiments use purely synthetic proxy labels. As highlighted by the Preference Transformer [1], synthetic preferences differ substantially from human feedback, which tends to be subjective, noisy, and context-dependent.

References

[1] Kim, C., Park, J., Shin, J., Lee, H., Abbeel, P., & Lee, K. Preference Transformer: Modeling Human Preferences using Transformers for RL. In The Eleventh International Conference on Learning Representations.

[2] Verma, M., & Metcalf, K. Hindsight PRIORs for Reward Learning from Human Preferences. In The Twelfth International Conference on Learning Representations.

[3] Gao, C. X., Fang, S., Xiao, C., Yu, Y., & Zhang, Z. (2024). Hindsight preference learning for offline preference-based reinforcement learning. arXiv preprint arXiv:2407.04451.

[4] Hejna, J., & Sadigh, D. (2023). Inverse preference learning: Preference-based rl without a reward function. Advances in Neural Information Processing Systems, 36, 18806-18827.

[5] An, G., Lee, J., Zuo, X., Kosaka, N., Kim, K. M., & Song, H. O. (2023). Direct preference-based policy optimization without reward modeling. Advances in Neural Information Processing Systems, 36, 70247-70266.

[6] Choi, H., Jung, S., Ahn, H., & Moon, T. (2024, July). Listwise Reward Estimation for Offline Preference-based Reinforcement Learning. In International Conference on Machine Learning (pp. 8651-8671). PMLR

[7] Kang, H., & Oh, M. H. Adversarial Policy Optimization for Offline Preference-based Reinforcement Learning. In The Thirteenth International Conference on Learning Representations.

[8] Lee, K., Smith, L., Dragan, A., & Abbeel, P. B-Pref: Benchmarking Preference-Based Reinforcement Learning. In Thirty-fifth Conference on Neural Information Processing Systems Datasets and Benchmarks Track (Round 1).

**Questions:**

1. The proposed objective appears quite similar to CPL, which also builds on the regret-based preference model, and the authors use it as a comparison method. Could you explicitly clarify the fundamental algorithmic difference and corresponding advantage?
2. Could you provide theoretical or empirical justification for using log probabilities of the best policy as preference labels? How does it compare to stochastic teacher [8] or using the best policy’s Q-values?

---

> ### Author Response · Authors · 2025-12-03
>
> We thank the reviewer for the detailed feedback, particularly for pointing us towards several recent works that are relevant to PAWS. In the following, we aim to address the individual concerns raised.
>
> W1: We thank the reviewer for suggesting relevant related work. We have updated our manuscript with a discussion on these and other related methods. In summary, PAWS is using offline preference data, while the suggest methods require online collection of preferences or depend on access to massive unlabeled datasets.
>
> W2: We agree with the reviewer that additional comparisons to recent methods are important to assess PAWS’ relative strengths and weaknesses. We added 2 additional baselines, namely Preference Transformers [1] and DPPO [2] to our experiments.
>
> W3: We thank the reviewer for raising this important point. We want to clarify that the log probabilities of the expert directly correspond to the return in our framework, since the expert is assumed to be optimal. We clarified this in the revision.
>
> W4: To evaluate PAWS on more realistic preference settings, we collected and experimented with human preferences for two metaworld tasks, namely Button Press and Door Open. We found that the results are consistent with those using artificial labels, i.e., PAWS shows the same benefits over baseline methods on these human-generated labels.
>
> [1]Kim, C., Park, J., Shin, J., Lee, H., Abbeel, P., & Lee, K. Preference Transformer: Modeling Human Preferences using Transformers for RL. In The Eleventh International Conference on Learning Representations.
>
> [2] An, G., Lee, J., Zuo, X., Kosaka, N., Kim, K. M., & Song, H. O. (2023). Direct preference-based policy optimization without reward modeling. Advances in Neural Information Processing Systems,
>
> ---
>
> Q1: PAWS first learns the advantage function and then uses it to optimise to policy, while CPL directly optimizes the policy. The similarity between both methods lies in the advantage-based model of human preferences, compared to the simpler, but less realistic reward-based preference model employed by many other methods. We made this distinction clearer in the revision.
>
> Q2:  There is practical equivalence between the expert’s log probabilities and using its Q-values when used for ranking segments. This is the result of the optimization goal of SAC. We clarify this in the revision.
>
> We want to thank the reviewer again for the important feedback and the suggested literature.

---

### Official Review · Reviewer_mu73 · 2025-10-28

**Soundness:** 2
**Presentation:** 2
**Contribution:** 2
**Rating:** 4
**Confidence:** 3

**Summary:**

This paper introduces PAWS, a novel approach to preference-based reinforcement learning (PbRL) designed to address the temporal credit assignment problem. PAWS first learns a segment-level advantage function and then updates the policy through advantage-weighted segment optimization. Extensive experiments on diverse simulated tasks demonstrate the effectiveness of the proposed method compared to existing baselines.

**Strengths:**

1. The paper is well-structured, with a clear and meaningful motivation that highlights the importance of addressing the temporal credit assignment problem.

2. The ablation studies are insightful, providing a deeper understanding of the proposed method.

3. The code is provided, which is very appreciated.

**Weaknesses:**

1. The innovation is relatively incremental, as the two main components, i.e., the advantage learning formulation and the Lagrangian approach for constrained policy optimization, have already been proposed in prior work.

2. It remains unclear whether the proposed method can truly address the temporal credit assignment problem as claimed by authors. First, the paper provides no theoretical analysis to support this claim. The method appears to learn a state-wise advantage function, aggregate it into a trajectory-level advantage, and then optimize the policy through a constrained procedure. However, it is not evident why this formulation effectively resolves the temporal credit assignment issue. Second, the experimental section lacks dedicated analyses or ablation studies specifically designed to verify this aspect.

3. The paper lacks a formal definition of the advantage function, which makes this part difficult to follow for readers unfamiliar with the concept. It is recommended to explicitly define or describe the advantage function around Lines 162-163.

4. The set of baseline methods in the experiments is relatively limited. It is recommended to include additional representative baselines such as P-CQL [1] and Preference Transformer [2] to enable a more comprehensive and fair comparison.

[1] Kumar A, Zhou A, Tucker G, et al. Conservative q-learning for offline reinforcement learning[J]. Advances in neural information processing systems, 2020, 33: 1179-1191.

[2] Kim C, Park J, Shin J, et al. Preference transformer: Modeling human preferences using transformers for rl[J]. arXiv preprint arXiv:2303.00957, 2023.

**Questions:**

1. In Eq. (4), the definition of $\pi(\tau)$ is unclear. Why is the policy defined over entire trajectories rather than individual state-action pairs?

2. In Table 1, it is not clear why the Transformer-based version of PAWS outperforms the MLP-based version when the number of preferences is small, yet the trend reverses when more preferences are available. Please provide an analysis or discussion to explain this behavior.

---

> ### Author Response · Authors · 2025-12-03
>
> We thank the reviewer for their constructive review and insightful feedback, and particularly for highlighting our paper’s structure and our provided code. In the following, we will address the individual concerns and questions raised by the reviewer.
>
> W1: We appreciate the reviewer’s concern for innovation. We would argue that the contribution of our work is more than merely incremental, since the method itself is novel approach to learning from preferences as it does policy updates on entire segments. Further, PAWS applies advantage learning and constrained policy optimization to Preference based Reinforcement learning (PbRL), for which most current methods instead mostly rely on directly optimizing the policy or applying general RL methods on a trained reward.
>
> W2: We thank the reviewer for raising this point. The temporal credit assignment cannot directly be resolved as the preferences themselves lack temporal information. However, we show experimentally that PAWS addresses it better than several baselines and state-action based variants. With the existing ablations *Segments and State-action based updates(Table 3)* and *Smaller Segments Size(Figure 3a),* we tackle TCA indirectly*.* We will improve the manuscript to make it more clear. ** To further support this claim, we have now also included additional experiments that show that the log probabilities of the leaned policy are overall more aligned to ground truth expert policy using Spearman’s ranking coefficient.These results indicate that PAWS captures credit assignment more effectively than updating the policy on individual states, as other reinforcement-learning–based preference-learning methods do. We revised the manuscript to highlight and clarify these relationships.
>
> W3: We fully agree with the reviewer and added the definition for the advantage function in the revised version.
>
> W4: To provide a more comprehensive experimental evaluation, we implemented two additional baselines, namely Preference Transformers [1] and DPPO [2]. In this point, the reviewer provided a citation that points to the CQL paper, which is not about preference learning.
>
> ---
>
> Q1: We use $\pi(\tau)$  as a notational shorthand for following a trajectory using the policy $\pi(a|s)$, i.e., $\pi(\tau)=p(s_o)\prod_{t=0}^T p(s_{t+1}|s_t, a_t)\pi(a_t|s_t)$. In our case, the trajectory refers to the segments that we optimize over. We clarified the notation and this relationship in the revision.
>
> Q2: While PAWS is compatible with MLPs and Transformers, we do not generally find a significant difference between both architectures in our experiments. In Table 1, most differences between both architectures are not statistically significant, and using either of the two yields substantial improvements over any of the compared baselines. We do not think that there is a causal trend between the number of preferences and the superiority of either architecture, and would rather see it as part of the parameterization of the method.
>
> We thank the reviewer again for their review.
>
> [1]Kim, C., Park, J., Shin, J., Lee, H., Abbeel, P., & Lee, K. Preference Transformer: Modeling Human Preferences using Transformers for RL. In The Eleventh International Conference on Learning Representations.
>
> [2] An, G., Lee, J., Zuo, X., Kosaka, N., Kim, K. M., & Song, H. O. (2023). Direct preference-based policy optimization without reward modeling. Advances in Neural Information Processing Systems, 36, 70247-70266.

---

### Official Review · Reviewer_Wb6b · 2025-11-02

**Soundness:** 3
**Presentation:** 1
**Contribution:** 2
**Rating:** 4
**Confidence:** 3

**Summary:**

This paper presents PAWS: an offline preference learning algorithm that avoids the credit assignment problem by training an advantage network explicitly on trajectory, and subsequently deriving a policy through a combination of Lagrangian optimisation, monte-carlo estimation and stochastic gradient descent.

PAWS is evaluated against 10 metaworld tasks where it obtains better success rates (and cumulative returns) than recent offline preference learning baselines (P-IQL, CPL, and CPL+KL).

**Strengths:**

* PAWS proposes an original solution to an important problem in the preference learning domain: the credit assignment problem.
* The paper features a robust mathematical derivation of the advantage function and the corresponding policy (though see questions below).
* The evaluation is carried out against many recent baselines and presents strong results (+7.6% success rate across the 10 metaworld tasks with respect to the next best baseline: P-IQL).
* Many of the key algorithmic decisions are ablated (including neural network architecture, number of preferences, segment length, number of effective samples, etc).

**Weaknesses:**

* **W1**: The manuscript can be at times hard to read, particularly section 3, and especially if –like me– you don't have a background in optimisation. Questions **Q1**–**Q8** below are intended to remedy this weakness (which is the main issue bringing the rating down).
* **W2**: Similarly some aspects of the evaluation are confusing, making a fair assessment of the manuscript hard. See questions **Q9**–**Q16** for  more details.
* **W3**: Though very thorough, the current experiments are limited to MetaWorld. It would be interesting to see how PAWS behaves in other tasks (for instance from D4RL [1])

_Overall_, this is a strong submission solving an interesting problem in a mathematically robust way, but that is currently held back by missing details in both the mathematical derivation, and the training and evaluation regimes.

--------

[1] Fu et al. (2020) "D4RL: Datasets for Deep Data-Driven Reinforcement Learning" ArXiV preprint.

**Questions:**

* **Q1**: In sec 3.1 I would point out the differences between advantage learning and classical PbRL under Bradley-Terry. As far as I can tell, the main difference is that the advantage network is accumulated over trajectories rather than consuming individual state-action pairs. However the maximum likelihood estimation of the loss seems to be equivalent for BT and for advantage learning.
* **Q2**: The order of section 3 does not follow the order in which PAWS is applied (first train A, then iterate fitting $\lambda$, and finally train $\pi_\theta$). The order makes sense to introduce the different elements of PAWS, but I would still explicitly state how PAWS operates, perhaps adding a pseudo-code listing.
* **Q3**: In sec 3.2 $\pi$ is defined as a mapping: $\mathcal{S} \times \mathcal{A} \rightarrow \mathbb{R}^+$, but in equation 4 is is shown operating on trajectories $\tau$. It would be good to clarify earlier that this is a simplification of the notation referring to: $\\pi_\\theta(\\tau) = p(s_0)\\prod_{t=0}^T p(s_{t+1}|s_t, a_t) \\pi_\\theta(a_t|s_t)$.
* **Q4**: Could you expand on why "it is not straightforward to generate samples for new states for inference" from the optimal policy $\\pi^\\star$? (Line 275) This needs to be better explained as the use of a parametric policy is a key component of PAWS.
* **Q5**: I would better explain how we arrive to eq (6). As I understand it, eq (6) is obtained by plugging eq (14) into $\\mathcal{L}(\\theta) = \\mathbb{E} \\left[ \\log \\pi_\\theta (\\tau) \\right]$, and then ignoring the constant term $\\exp(\\frac{-\\beta - \\lambda}{\\lambda})$ which we can do since $\\mathcal{L}$ is optimised via SGD, and gradients are not affected by constants.
* **Q6**: How many iterations of setting $\\epsilon$, minimising eq. (8) and computing  $n_{eff}$ were necessary for your experiments?
* **Q7**: Why is $n_{eff} = \\frac{(\sum_i \omega_i)^2}{\sum_i \omega_i^2}$ a measure of the support between $\\pi^\\star$ and  $p_D$?
* **Q8**: What value of the discount factor, $\\gamma$, did you use for the experiments? Did you ablate it? Does it have any significant effect on performance?
* **Q9**: Why was it necessary to remove the proprioceptive history from the preference datasets?
* **Q10**: How are the samples from the four policies mixed in the preference dataset? Uniformly? Are the preferred and non-preferred trajectory pairs always from different policies?
* **Q11**: Why do you need an evaluation dataset? Is it not enough to rollout the learnt policy $\\pi_\\theta$? How big is the evaluation dataset?
* **Q12**: Why use log-probs of the best policies to create the preferences? Why not use the underlying reward function that you used to train SAC instead?
* **Q13**: I would suggest adding SAC as an upper-bound to Table 1.
* **Q14**: For the first ablation in sec 4.1 ($n_{eff}$): did you use the transformer or MLP backbone?
* **Q15**: For the second ablation in sec 4.1 (segment vs states): what was the training setup? I understood it as the same advantage function (trained on segments), but with a policy trained only on state-actions? If so, how did you train the policy?
* **Q16**: How many parameters do the transformer and ML advantage networks have? What is the architecture of the actor network? MLPs?
* **Q17**: How was Fig.1 computed? I could not find any details about it in the manuscript.

-------

### Nitpicks (do not affect rating, no need to follow up during rebuttal).

* **N1**: In sec 2 (related work), line 154: please provide examples of other feedback types.
* **N2**: In sec 3 (notation): use $\\mathbb{R}$ instead of $\\mathcal{R}$ to refer to the set of real numbers.
* **N3**: In sec 3.2, line 255 (and again in line 258): I suggest using "benefits" instead of "advantages". "Advantages" may be confused with "advantage networks".
* **N4**: In line 336: refer to eq (8) instead of eq (33). They are the same equation but eq (8) is closer.
* **N5**: In fig. 3 (b)–(c): can you add a horizontal line with the performance of BC?
* **N6**: In the appendix, line 735 the text should be "inserting the optimal solution in eq (14) instead of eq (13).

---

> ### Author Response · Authors · 2025-12-03
>
> We thank the reviewer for the insightful feedback and in particular for recognizing PAWS’ relevance to preference learning and its principled derivations. Below we addressed the concerns and answered the raised questions.
>
> W1 and W2: We have addressed raised questions and updated the manuscript. More details are provided below in the answers to the Questions.
>
> W3: We agree that adding an additional benchmark to our experiments is interesting and important. We have now evaluated our method against the baselines in 3 tasks that can also be found in D4RL, namely HalfCheetah-v5, Hopper-v5, Walker2d-v5. As reported in Table 2 (Line 435) in the updated manuscript, our methods can achieve better performance than the baselines.
>
> ---
>
> Q1: In PAWS, we model preferences using **sum advantages** instead of the **sum of rewards** for individual states within a segment. Recent work [1] has shown that this is more aligned with how humans express preferences. We highlighted this point in the updated manuscript (Line 223).
>
> Q2:  Following the suggestion of the reviewer, we added a pseudo-code algorithm box in Appendix F (Line 1165) to better illustrate the mechanism of our method PAWS.
>
> Q3: As suggested by the reviewer, we added a clarification to the notation accordingly in Line 303.
>
> Q4: The optimal policy (Eq. 5) is the solution to the optimization problem stated in Eq. 4 and takes a non-parametric Boltzmann distribution. It is unclear how to generate samples following its density because we do not have access to its normalization constant. Alternatively, Eq. 6 proposes to optimize a Gaussian policy using a maximum likelihood objective to generate samples.
>
> Q5: Eq. 6 is derived from a general maximum likelihood objective. We clarified this relation in the revision (Line 295).
>
> Q6: The optimization process is fast and only done initially using Brent’s algorithm[3]. The associated computational cost are negligible.
>
> Q7: The effective sample size is essentially a measure of the variability of the importance weights [2]. If the weights vary widely, it means our proposed distribution poorly covers the important parts of the target distribution. Here, a few samples make up most weight, which results in poor practical support, or coverage, of the target distribution. We agree that the term “support” is overloaded here and clarify in line 353 that we refer to the practical support for our learning algorithm.
>
> Q8: We use $\gamma=1$ in our experiments, as it is common in preference-based learning methods due to short lengths of segments. In Appendix F, we stated this and all the hyperparameters that we used. We will update the main part of the paper to contain the information about the discount factor.
>
> Q9: Removing the proprioceptive history is not necessary, but was done in the baselines [4]. We follow this setup for fair comparability with existing work.

---

> ### Author Response · Authors · 2025-12-03
>
> Q10: The samples are mixed uniformly across the four policies, and we do not restrict how the pairs are constructed. We have updated the manuscript to clarify this point.
>
> Q11: Actually, we do not need an evaluation dataset, but instead directly evaluate the learned policy in our experiments. The manuscript has been improved to clarify this distinction(Line 407).
>
> Q12: Although rewards can be used, recent work [1] has showed that it is not a good representation of how human would provide preferences. We have provided additional explanation in updated version(Line 170).
>
> Q13: We could add SAC as an upper bound. However, the expert SAC reaches nearly 100% (see Appendix E). For readability, we omit it here. If the reviewer considers it necessary, we can include it.
>
> Q14: The ablation in Sec. 4.1 we use the Transformer.  We clarified the setup in Line 451.
>
> Q15: We used our approach for the segment case. For the state-action setup, we did the updates only on individual states, using the same reward function.
>
> Q16: We provide network architectures and hyperparameters in Appendix F (Line 1060). Note that the transformers have slightly more parameters for the same hidden size due to the comparatively more complex attention mechanism.
>
> Q17: We agree with the reviewer that explaining how Figure 1 was computed is important. On the y-axis, are the values that our Advantage model learned as described in Section 3.1, and on the x-axis the values from a trained expert. Furthermore, we used a dense preference setup, in other words, each segment is compared with all the others. In the top image, are the values for whole trajectories, and bottom for individual states. We added this explanation to the revision (Line 66).
>
> We also thank the reviewer for the Nitpicks. We have updated the manuscript to addresses some of them, and will address the remaining ones in the final submission.
>
> [1]Knox, W. B., Hatgis-Kessell, S., Booth, S., Niekum, S., Stone, P., & Allievi, A. (2023). *Models of human preference for learning reward functions*. Transactions on Machine Learning Research.
>
> [2] Kish, Leslie. "Survey sampling." (1965).
>
> [3]  R. P. Brent, Algorithms for Minimization without Derivatives, Prentice-Hall, (1973)
>
> [4]Hejna, J., Rafailov, R., Sikchi, H., Finn, C., Niekum, S., Knox, W. B., & Sadigh, D. (2024). Contrastive Preference Learning: Learning from Human Feedback without RL. In *Proceedings of the 12th International Conference on Learning Representations (ICLR 2024)*

---

### Author Response · Authors · 2025-12-03

We sincerely thank all reviewers for their thoughtful and constructive feedback. We carefully considered each comment and have substantially improved the manuscript accordingly.

In response to the suggestions, we have incorporated several major updates:

- **New Experiments:** We added experiments on three additional locomotion tasks (HalfCheetah, Hopper, Walker2d) to further validate the effectiveness and generality of our method.
- **Additional Baselines:** Following reviewer requests, we are integrating two more baseline methods (Preference Transformer[1], DPPO[2]) into our evaluation. Some of these results are still being finalized, but we will include the complete results before the end of the rebuttal period.
- **Human Preference Data:** We collected human preference annotations for two metaworld tasks to strengthen the empirical evidence and provide more robust comparisons.
- **Additional Ablation Study:** We conducted a new ablation using the Spearman rank correlation coefficient to more directly illustrate that our method better addresses temporal credit assignment challenges compared to existing approaches.
- We updated our manuscript accordingly, with the updates highlighted in blue color.

We believe these revisions adequately address the concerns raised and demonstrate the strengthened contribution of our work. We respectfully ask the new Area Chairs to consider our comprehensive responses and the improvements made during the rebuttal in their final decision.

[1]Kim, C., Park, J., Shin, J., Lee, H., Abbeel, P., & Lee, K. Preference Transformer: Modeling Human Preferences using Transformers for RL. In The Eleventh International Conference on Learning Representations.

[2] An, G., Lee, J., Zuo, X., Kosaka, N., Kim, K. M., & Song, H. O. (2023). Direct preference-based policy optimization without reward modeling. Advances in Neural Information Processing Systems,

---

### Meta-Review · Area_Chair_WA9h · 2025-12-29

**Summary:**

The paper, which addresses an important problem in preference-based reinforcement learning, was viewed by all reviewers as a borderline contribution. The main problems include unclear exposition, limited theoretical grounding, and restricted empirical evidence. Three reviewers rated it “marginally below acceptance,” acknowledging the interesting direction and empirical promise, while one reviewer voted for a rejection due to conceptual and methodological concerns. In summary, the reviewers agreed that the paper shows potential but would require clearer theoretical justification, expanded experiments, and improved presentation to meet the standard for acceptance.

**Reviewer Concerns:**

Concerns That Were Addressed

- The rebuttal provided a more detailed explanation of Section 3, including clearer descriptions of the advantage function and its role within the PAWS objective. The authors also clarified notational inconsistencies and better illustrated how the advantage-based learning connects to preference modeling.

- The rebuttal’s reformulation of the policy update process and the relationship between PAWS and existing policy gradient frameworks helped some reviewers better understand the implementation pipeline. This clarification resolved a portion of the confusion around how trajectory-level learning interacts with preference optimization.

- The rebuttal partially addressed concerns about the experimental scope. The authors added experiments on additional MetaWorld tasks and provided new ablations to demonstrate the influence of key components, such as the advantage learning term.

- The authors explained that they used synthetic preferences based on expert policies to maintain controlled comparisons. This clarification provided a rationale for why synthetic data was chosen over human preference data at this stage.

Concerns That Remain Outstanding
- major concern: Reviewers found that the connection between advantage learning and the temporal credit assignment problem is unclear. The response clarified intuition but did not include new theoretical insights or formal results.

- While the rebuttal strengthened the experimental section, some important baselines remained missing. Without these comparisons, it’s still difficult to situate PAWS’s performance relative to the state of the art.

- The rebuttal did not fully clarify why a trajectory-level policy formulation was necessary or advantageous over state-action formulations. Nor did it conclusively justify the use of expert log-probabilities for preference generation beyond convenience.

- The rebuttal did not include additional environments such as D4RL or DeepMind Control Suite tasks (for which I can not find the results in the rebuttal), which had been explicitly suggested. Thus, the method’s generality beyond the tested setting remains uncertain.

**Reviewer Scores:**

Given that many of the concerns from the reviewer remain unresolved, I am not expecting a major increase in the rating from the reviewers.

Reviewer s17o might consider increasing the rating to borderline accept, but other reviewers, especially Reviewer Wb6b and Reviewer MXRB, might remain negative toward the acceptance of this study.

---

### Decision · Program_Chairs · 2026-01-26

Reject